# GWAS for primary angle-closure glaucoma identifies loci related to ocular biometry and morphology

GWAS of primary angle-closure glaucoma have identified eight loci conferring risk in Asian populations. However, it remains unclear whether the genetic risk factors for the disease are consistent across different populations. Here, we present a discovery GWAS for primary angle-closure glaucoma in Europeans using the UK Biobank. We replicate our findings in six independent European populations and compare these results with results from 14 Asian cohorts. Five genomic regions in the discovery cohort are associated at genome-wide significance, including two loci previously identified in Asian cohorts. We next meta-analyse the discovery and replication cohorts to identify six additional novel loci, all previously associated with refractive error. Mendelian randomisation provides evidence for a causal role of shorter axial length and hypermetropic refractive error on primary angle-closure glaucoma. A polygenic risk score derived from the European ancestry meta-analysis demonstrates significant associations with quantitative ocular traits – including a shallower anterior chamber and higher intraocular pressure – in the independent EPIC-Norfolk cohort. Finally, a multi-ancestry meta-analysis of all 21 European and Asian cohorts identifies 12 further novel loci. This work shows that genetic factors associated with a darker iris and hypermetropia confer risk for primary angle-closure glaucoma.

Primary angle-closure glaucoma (PACG) is an important cause of preventable blindness estimated to affect more than 20 million people worldwide in 2020[1]. While less common than primary open-angle glaucoma (POAG), PACG is three times more likely to result in blindness[2], making it responsible for a significant proportion of all glaucoma-related visual impairment[3]. PACG arises when the anterior chamber drainage angle of the eye, situated between the iris root and peripheral cornea, is closed (i.e., contact between the iris and the trabecular meshwork), impeding the egress of aqueous humour from the eye, leading to an elevation of intraocular pressure (IOP) and consequent glaucomatous optic neuropathy.

Tornquist and Alsbirk proposed genetic control of ocular dimension influencing risk of PACG more than 50 years ago[4,5]. More recently, evidence of PACG heritability has become well-established[6,7],

however the number of biologically relevant loci discovered from genome-wide association studies (GWAS) has been modest[8,9] with implicated pathways including cell–cell adhesion, collagen metabolism and acetylcholine metabolism.

Three-quarters of the global burden of PACG is estimated to occur in Asia[1] and previous discovery GWAS[8,9] have been derived largely from Asian cohorts. However, an estimated 2.6 million people of European ancestry have PACG[10], representing a significant disease burden, and it remains unclear whether the genetic associations of PACG might be different in Europeans, given the contrasting epidemiology.

Here we report a large-scale GWAS for PACG in Europeans using a European discovery cohort, UK Biobank (UKB), with replication in six independent European populations. We then compare and combine this genetic discovery with results from 14 predominantly Asian

e-mail: r.luben@ucl.ac.uk

cohorts and probe plausible biological mechanisms using Mendelian randomisation. Lastly, we construct polygenic risk scores (PRS) and test associations with quantitative ocular traits and PACG status in the independent EPIC-Norfolk Eye Study[11].

## Results and discussion

Figure 1 summarises the study design, participant numbers and exclusions in the various phases of the study.

### Discovery GWAS

The primary discovery GWAS was carried out in UK Biobank (1564 cases, 439,185 controls). Five genomic regions associated with PACG were identified at genome-wide significance ($P < 5 \times 10^{-8}$) with the most significant association at a locus in the region of *HERC2* and *OCA2* (rs12913832:G, OR 1.29, $P = 5.62 \times 10^{-10}$). These genes are central to melanin synthesis[12] and consistently associate with a variety of pigment-related traits, including skin, hair, and iris colour[13,14]. Another novel PACG-associated locus at *SEMA3A* (rs17245595:C, OR 1.22, $P = 3.02 \times 10^{-8}$) has previously been associated with morphological characteristics and colour of the iris[15,16]. Biologically, iris colour is related to the degree of pigmentation present, with greater levels of melanin resulting in a darker and thicker iris[17]. Iris thickness is thought to play a mechanistic role in the development of angle-closure and PACG[18,19], and the relative homogeneity of eye colour in Asian populations may explain the absence of these findings in previous studies. The relationship between lighter coloured eyes and crypts and furrows is well-established. Several studies have reported associations between iris surface features, lighter eye colour, and iris thickness in Asian populations suggestive of a link with angle closure[20,21]. However, previous GWAS of PACG were not able to detect loci related to iris crypts.

We also observed a genome-wide significant association with PACG at a new locus *PCDH7* (rs184176302:C, OR 0.44, $P = 1.06 \times 10^{-8}$) and confirm the significant associations at *PXDNL* (rs11984688:A, OR 0.51, $P = 1.11 \times 10^{-9}$) and *GLIS3* (on chromosome 8, near PCMTD1 – ST18; rs746970:G, OR 1.24, $P = 2.29 \times 10^{-8}$), both previously associated with PACG in Asian populations. *GLIS3* has also been associated with congenital glaucoma and with POAG[22,23].

We then examined the association of the lead variant at each of our PACG-associated loci from the discovery phase in each of the six independent replication cohorts. Supplementary Table 1 summarises the associations found at discovery and a combined estimate for replication. All loci identified at discovery were at least nominally significant ($P < 0.05$) at replication, except for the locus near *PCDH7*, and all were directionally consistent. Supplementary Fig. 1 shows forest plots for the loci in the discovery cohort, each of the replication cohorts, and an overall estimate of the replication. Associations were largely driven by the FinnGen, UK, and Australian cohorts and associations for the loci at *PXDNL*, *HERC2* and *GLIS3* were individually significant ($P < 0.05$) in these cohorts.

### European ancestry meta-analysis

To increase power to detect PACG associations, we then meta-analysed all European cohorts, finding a total of ten genome-wide significant loci. Table 1 shows associations for the meta-analysis. Regional association analysis was performed to visualise the loci identified and illustrate linkage disequilibrium between variants within each locus (Supplementary Fig. 2). All loci identified in phase 1 discovery, except for *PCDH7*, remained genome-wide significant and a further six novel loci were identified (Supplementary Fig. 3). These include loci previously associated with a more hypermetropic refractive error or

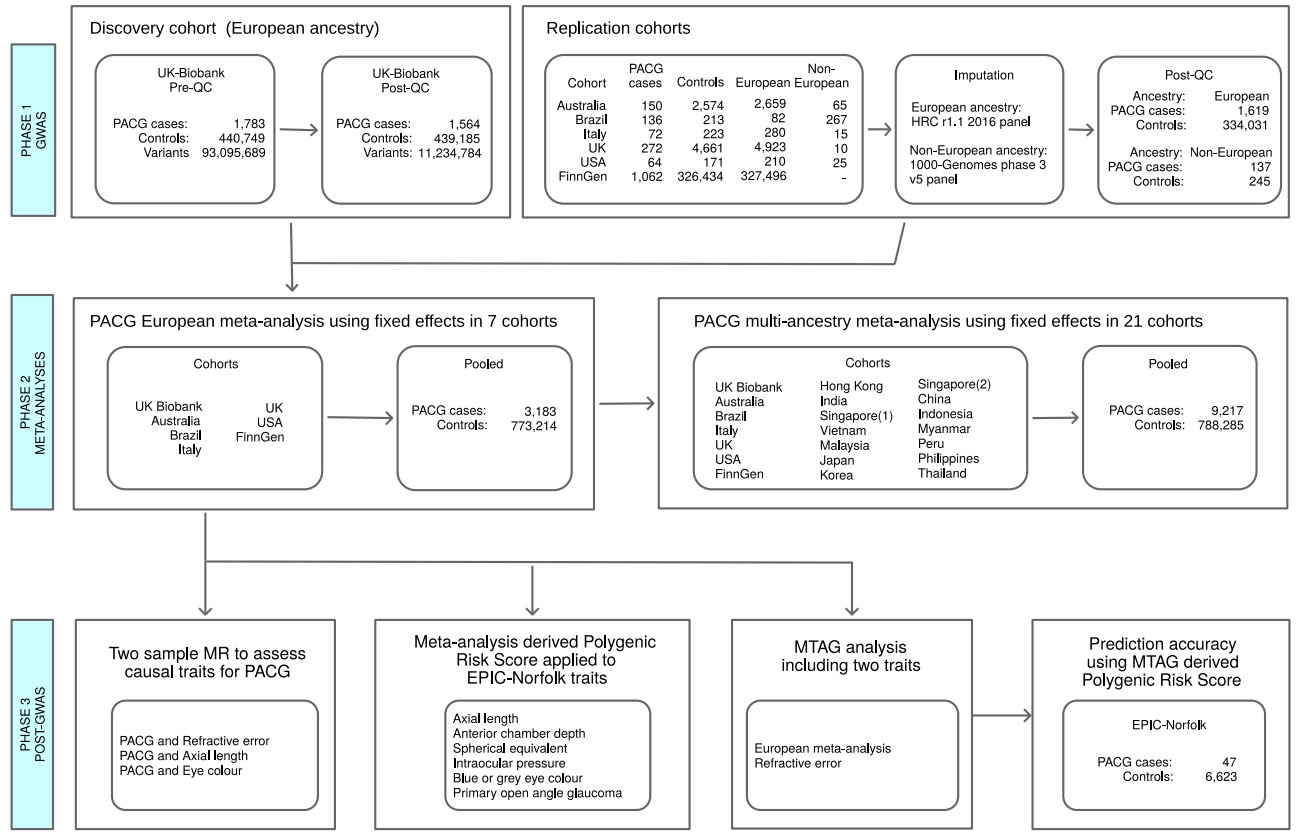

**Fig. 1 | Study design.** Overview of study design showing the different phases of our study, the cohorts and summary statistics used, and analyses undertaken. GWAS genome-wide association study, QC quality control, PACG primary angle-closure glaucoma, HRC haplotype reference consortium, MTAG multi-trait analysis of genome-wide association study, MR Mendelian randomisation.

**Table 1 | Lead variants from the ten genomic regions associated with PACG in the European ancestry meta-analysis (3183 cases, 773,214 controls)**

| Variant | Chromosome: position | Designation | Nearest gene | Reference: effect allele | Effect allele frequency | Imputation information score | Odds ratio per allele | P value | Ocular biometry or refraction | Iris colour or morphology |
|---|---|---|---|---|---|---|---|---|---|---|
| rs10215636 | 7:83571278 | Intergenic | SEMA3A | C:T | 0.62 | 1.00 | 1.18 | $9.83 \times 10^{-10}$ | | ✓ |
| rs10818834 | 9:126317324 | Intron | DENND1A | C:T | 0.71 | 0.99 | 1.17 | $4.56 \times 10^{-8}$ | ✓ | |
| rs111736202 | 8:58787896 | Intergenic | FAM110B | A:G | 0.98 | 0.98 | 0.64 | $3.11 \times 10^{-8}$ | | |
| **rs11984688** | **8:52647065** | **Intron** | **PXDNL** | **A:C** | **0.98** | **0.99** | **0.52** | **$1.02 \times 10^{-15}$** | | |
| rs12193446 | 6:129820038 | Intron | LAMA2 | G:A | 0.90 | 1.00 | 0.78 | $1.28 \times 10^{-9}$ | ✓ | |
| rs12913832 | 15:28365618 | Intron | HERC2 | G:A | 0.19 | 1.00 | 1.31 | $7.93 \times 10^{-18}$ | ✓ | ✓ |
| rs142455016 | 7:28255376 | Intron | JAZF1 | T:C | 0.99 | 0.85 | 0.43 | $7.02 \times 10^{-9}$ | | |
| rs73199928 | 4:6875213 | Intron | KIAA0232 | A:G | 0.98 | 0.91 | 0.61 | $3.57 \times 10^{-8}$ | | |
| **rs746970** | **9:4217822** | **Intron** | **GLIS3** | **G:A** | **0.66** | **1.00** | **1.21** | **$7.22 \times 10^{-12}$** | ✓ | |
| rs7744813 | 6:73643289 | Intron | KCNQ5 | A:C | 0.40 | 0.98 | 1.16 | $1.73 \times 10^{-8}$ | ✓ | |

Results are presented for the European primary angle-closure glaucoma (PACG) genome-wide association study meta-analysis (UK Biobank discovery plus six replication cohorts). GWAS associations used logistic regression with a significance level of $P < 5 \times 10^{-8}$ to account for multiple comparisons, and the Wald test statistic with two-sided testing. The presence of a tick in the last two columns (under Ocular biometry or refraction and Iris colour or morphology) indicates associations at genome-wide significance ($P < 5 \times 10^{-8}$) previously reported in the published literature and described in the Discovery GWAS section of the main text. Loci previously associated with PACG are shown in bold.

smaller axial dimensions, such as *LAMA2* (rs12193446:A, OR 0.78, $P = 1.28 \times 10^{-9}$), *KCNQ5* (rs7744813:C, OR 1.16, $P = 1.73 \times 10^{-8}$) and *DENND1A* (rs10818834:T, OR 1.17, $P = 4.56 \times 10^{-8}$), supporting the known clinical association between small eye phenotypes and PACG[24]. This contrasts with a previous study which failed to find a significant PACG association in an Asian cohort for loci previously associated with axial length[8]. If axial length was related to PACG in Asian populations, the high prevalence of myopia in many Asian countries[25] compared to Europeans, might suggest a lower rate of PACG. High rates of both PACG and myopia in Asians may imply that any putative causal association between axial length and PACG is restricted to Europeans; however, this is contentious for several reasons. Rapid industrialisation in many Asian countries has accompanied pronounced changes to ocular biometric characteristics[26] with a secular trend of increasing myopia in younger people[27,28].

## Multi-ancestry analysis

Table 2 and Fig. 2 contrast our European ancestry meta-analysis with published GWAS results from Asian cohorts. Khor and colleagues examined >10,000 PACG cases and identified disease association at five loci and reconfirmed three previously reported loci[8]; genome-wide significant loci at *COL11A1*, *EPDR1* and *FERMT2* were strongly associated and directionally consistent in our European ancestry meta-analysis ($P < 10^{-5}$), while loci at *FNDC3B* and *FAM102A* were non-significant in Europeans. Our findings support the established clinical relationships of PACG including hypermetropia, however, the most significant loci, except at *PXDNL* and *GLIS3*, were not genome-wide significant in the previous Asian GWAS. Genome-wide significant loci in Europeans were directionally consistent in Asians, except at *HERC2*.

To further increase power and given directionally consistent results that suggest there is not substantial heterogeneity (Supplementary Fig. 4), a multi-ancestry meta-analysis was performed combining our European ancestry meta-analysis with 14 cohorts (Fig. 1) from Asian countries, including China, Vietnam, Thailand, Malaysia, Singapore, Japan and India. An additional 12 novel loci reached genome-wide significance in the multi-ancestry analysis that were not independently significant in the Asian or European ancestry meta-analyses (Supplementary Table 3 and Supplementary Fig. 5), with most just below genome-wide significance in either the European or Asian populations. Supplementary Table 3 shows the significance

levels categories for each of the multi-ancestry in the European ancestry and Asian ancestry meta-analyses. Multi-ancestry associations were directionally consistent to both the European ancestry and Asian ancestry meta-analyses but loci at *BBS9* and *FAM102A* were driven by a strong association in Asians while those near *SRFBP1* and *DENND1A* had a stronger association in Europeans. Supplementary Table 4 shows reported associations for ocular biometry or refraction, iris colour or morphology, and any form of glaucoma in multi-ancestry loci. Novel loci near *SH3YL1*, *SOX2-OT* and *FBXO7* have been previously associated with ocular biometry or refraction[29,30] while loci near *BBS9*, *ARHGEF12* and *TTC28* were previously associated with POAG[31]. Notably, the *ARHGEF12* locus mapped to previous associations with IOP[32–34], POAG, and exfoliation syndrome[35], suggesting possible pleiotropy and shared pathogenesis pathways for PACG with other common causes of glaucoma. However, there were no PACG-related clinical relationships for the other loci.

## Genetic correlations

We examined genetic correlations between our European ancestry meta-analysis results for PACG and traits associated with ocular biometry, refraction and iris colour and morphology, using linkage disequilibrium (LD) score regression[36] (Supplementary Fig. 6 and Supplementary Table 5). This exploratory analytical approach, using summary statistics from publicly available sources, provides a broad overview of phenotypes suspected of being related to PACG. Myopic refractive error ($r = -0.41$, $P = 1.21 \times 10^{-27}$), longer axial length ($r = -0.50$, $P = 9.04 \times 10^{-9}$) and prior cataract surgery ($r = -0.17$, $P = 1.91 \times 10^{-3}$) were negatively correlated while hypermetropia ($r = 0.45$, $P = 2.94 \times 10^{-5}$) and the age when glasses started to be worn ($r = 0.24$, $P = 3.58 \times 10^{-7}$) were positively correlated. The negative correlation between PACG and IOP ($r = -0.14$, $P = 1.96 \times 10^{-3}$) appears counterintuitive. While PACG is clinically associated with high IOP, it may be that the association of lower IOP in shorter eyes in the general population is stronger than the association of higher IOP which only manifests in a subset of eyes at risk of PACG[8]. Furthermore, since IOP in PACG is highly labile, high IOP will only be detected, given current screening protocols, in small numbers of people at an advanced stage in the disease. The pigmentation-related traits (skin colour and hair colour), anthropometry-related traits (standing height and waist circumference) and education did not have a significant genetic correlation with PACG (Supplementary Table 5).

**Table 2 | Comparison of genome-wide significant loci in European and Asian populations**

| | Asian ancestry | | | | | | European ancestry | | | | | | | |
|---|---|---|---|---|---|---|---|---|---|---|---|---|---|---|
| | Variant | Chromosome: position | Ref: effect allele | MAF | Odds ratio per allele | P value | Variant | Chromosome: position | Ref: effect allele | MAF | Odds ratio per allele | P value | Nearest gene | LD ($R^2$) |
| **A** | | | | | | | | | | | | | | |
| | rs6446548 | 4:6903159 | A:G | 0.21 | 0.95 | 0.17 | rs73199928 | 4:6875213 | A:G | 0.01 | 0.61 | $3.57 \times 10^{-8}$ | KIAA0232 | 0.07 |
| | rs951762 | 6:73626639 | A:C | 0.36 | 1.13 | 0.0018 | rs7744813 | 6:73643289 | A:C | 0.39 | 1.16 | $1.73 \times 10^{-8}$ | KCNQ5 | 0.85 |
| | rs12205363 | 6:129834629 | T:C | 0.01 | 0.87 | 0.33 | rs12193446 | 6:129820038 | G:A | 0.10 | 0.78 | $1.28 \times 10^{-9}$ | LAMA2 | 0.60 |
| | rs849314 | 7:28282062 | A:G | 0.14 | 1.02 | 0.76 | rs142455016 | 7:28255376 | T:C | 0.01 | 0.43 | $7.02 \times 10^{-9}$ | JAZF1 | 0.03 |
| | rs10215636 | 7:83571278 | C:T | 0.06 | 0.99 | 0.82 | rs10215636 | 7:83571278 | C:T | 0.40 | 1.18 | $9.83 \times 10^{-10}$ | SEMA3A | 1.00 |
| | rs10504248 | 8:58789168 | T:C | 0.03 | 0.80 | 0.033 | rs111736202 | 8:58787896 | A:G | 0.02 | 0.64 | $3.11 \times 10^{-8}$ | FAM110B | 0.39 |
| | rs6478623 | 9:126315123 | T:G | 0.28 | 1.09 | 0.031 | rs10818834 | 9:126317324 | C:T | 0.28 | 1.17 | $4.56 \times 10^{-8}$ | DENND1A | 0.97 |
| | rs12913832 | 15:28365618 | G:A | 0.04 | 0.85 | 0.035 | rs12913832 | 15:28365618 | G:A | 0.36 | 1.31 | $7.93 \times 10^{-18}$ | HERC2 | 1.00 |
| **B** | | | | | | | | | | | | | | |
| | rs3753841 | 1:103379918 | G:A | 0.35 | 1.18 | $1.18 \times 10^{-11}$ | rs3753841 | 1:103379918 | G:A | 0.39 | 1.14 | $6.30 \times 10^{-7}$ | COL11A1 | 1.00 |
| | rs3816415 | 7:37988311 | A:G | 0.11 | 1.28 | $1.28 \times 10^{-12}$ | rs3816415 | 7:37988311 | A:G | 0.14 | 1.19 | $5.35 \times 10^{-6}$ | EPDR1 | 1.00 |
| | rs3739821 | 9:130702477 | A:G | 0.35 | 0.86 | $7.08 \times 10^{-10}$ | rs3739821 | 9:130702477 | A:G | 0.24 | 1.00 | 0.88 | FAM102A | 1.00 |
| | rs1258267 | 10:50895770 | G:A | 0.15 | 0.80 | $5.06 \times 10^{-14}$ | rs1153796 | 10:50901671 | G:A | 0.09 | 0.88 | $6.95 \times 10^{-3}$ | CHAT | 0.39 |
| | rs11024102 | 11:17008605 | G:A | 0.35 | 1.20 | $4.10 \times 10^{-15}$ | rs7123383 | 11:17002985 | A:G | 0.25 | 1.10 | $5.17 \times 10^{-4}$ | PLEKHA7 | 1.00 |
| | rs7494379 | 14:53411391 | T:C | 0.37 | 0.85 | $9.26 \times 10^{-9}$ | rs7494379 | 14:53411391 | T:C | 0.32 | 0.89 | $1.67 \times 10^{-5}$ | FERMT2 | 1.00 |
| **C** | | | | | | | | | | | | | | |
| | rs1015213 | 8:52887541 | G:A | 0.06 | 0.69 | $1.75 \times 10^{-9}$ | rs11984688 | 8:52647065 | A:C | 0.02 | 0.52 | $1.02 \times 10^{-15}$ | PXDNL | 0.16 |
| | rs736893 | 9:4217028 | G:A | 0.27 | 1.16 | $4.23 \times 10^{-8}$ | rs746970 | 9:4217822 | G:A | 0.33 | 1.21 | $7.22 \times 10^{-12}$ | GLIS3 | 0.96 |

LD linkage disequilibrium.

Genome-wide significant PACG loci grouped by [A] significant loci from the European ancestry meta-analysis in this study; [B] significant loci from published results of Asian populations (Khor et al.[9]); [C] significant loci in both European and Asian ancestries from this study and published results. GWAS associations used logistic regression with significance level $P < 5 \times 10^{-8}$ to account for multiple comparisons, and the Wald test statistic with two-sided testing.

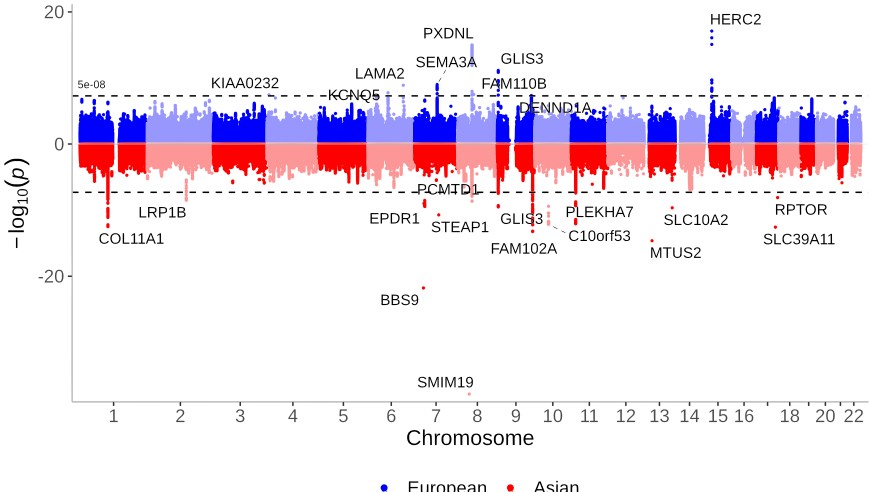

**Fig. 2 | Mirrored Manhattan plots presenting genetic associations with PACG in Europeans (upper) and Asians (lower).** Manhattan plots comparing European and Asian associations. The lower plot shows PACG associations by genomic position in a meta-analysis of 14 Asian cohorts. The upper plot shows PACG associations by genomic position in the European meta-analysis of seven cohorts. GWAS associations used logistic regression with significance level $P < 5 \times 10^{-8}$ (shown as a blue dotted line) to account for multiple comparisons, and the Wald test statistic with two-sided testing.

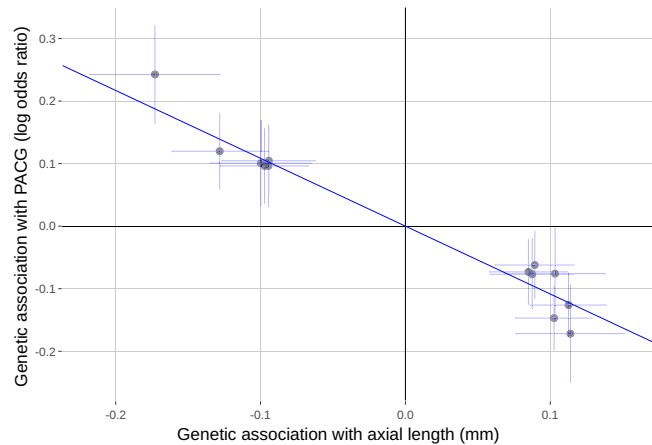

**Fig. 3 | Results for the Mendelian randomisation experiment examining the potential causal effect of axial length on PACG.** Two-sample inverse-variance weighted Mendelian randomisation using 13 independent genome-wide significant axial length loci as exposure instrumental variables and European meta-analysis PACG summary statistics as outcome. Data are presented as log odds ratios for genetic associations of PACG and the corresponding genetic association of axial length in mm for each locus. Error bars represent 95% confidence intervals. All units are per risk allele.

## Mendelian randomisation

Given the possible role of eye colour we have identified and the previously reported associations of PACG with hypermetropic refractive error and smaller axial dimensions, we used two-sample Mendelian randomisation (MR) to examine whether these relationships with PACG are causal (Supplementary Table 6). There was strong evidence for both shorter axial length (OR 0.34 (95% CI 0.29–0.40, $P < 0.001$) per mm increase in axial length, Fig. 3) and more hypermetropic refractive error (OR 2.99 (95% CI 2.58–3.47, $P < 0.001$) per unit increase in refractive error Z-score, Supplementary Fig. 7) causing PACG. While the overall MR results appeared to support a causal relationship between darker eye colour and PACG (Supplementary Fig. 8), this appears to be driven by a single strong effect variant (at the *HERC2* locus, which we identified in our European meta-analysis analysis) as evidenced by a leave-one-out analysis (Supplementary Fig. 9). This

suggests that darker eye colour may not cause PACG and that the *HERC2* locus influences risk by a different mechanism (pleiotropy). Notably, this variant (rs1129038) is also strongly associated with refractive error[29] and may mediate PACG risk through this pathway. Alternatively, only a subset of pathways determining eye colour may impact PACG risk, and these may be upstream of eye colour in the causal pathway. For example, eye colour may be a surrogate for iris thickness.

## Multi-trait analysis of GWAS

We aimed to leverage the strong evidence of genetic correlation and causality for refractive error on PACG to increase the power for genetic discovery for PACG, using the Multi-Trait Analysis for GWAS (MTAG) approach[37]. MTAG analysis included our current PACG European ancestry meta-analysis results and previously published refractive error summary statistics from a GWAS of 542,934 European participants[29]. A conditional analysis on the MTAG results identified 134 independent loci associated with PACG (Supplementary Data 1) of which 124 were genome-wide significant in the MTAG. A literature search using Open Targets Genetics[38] searching by variant and gene showed that 122 loci were known to be associated with ocular biometry or refraction (84 at genome-wide significance level), 23 loci were associated with iris colour or morphology (4 at genome-wide significance) and 46 associated with any form of glaucoma (3 at genome-wide significance). The loci at *KCNQ5*, *LAMA2*, *PXDNL* and *OCA2-HERC2* were genome-wide significant in both the MTAG analysis and the European ancestry meta-analysis.

## Population-level prediction performance using PRS

We constructed PRSs using SBayesRC, a Bayesian method that weights variants according to both the magnitude of their effect size on PACG and their prior probability of being a variant with functional effects[39]. We first created a PRS (PRS-A) using the MTAG analysis to maximise PACG discovery power, and this was tested on PACG ascertained in the independent EPIC-Norfolk Eye Study. In Fig. 4, we examined whether PRS-A could predict PACG or primary angle closure (PAC) cases status compared to controls (defined as study participants excluding those with other forms of glaucoma and suspected glaucoma). The area under the receiver operating characteristic curve (AUROC) for a model including PRS-A, age and sex was 0.75, while a model including age and

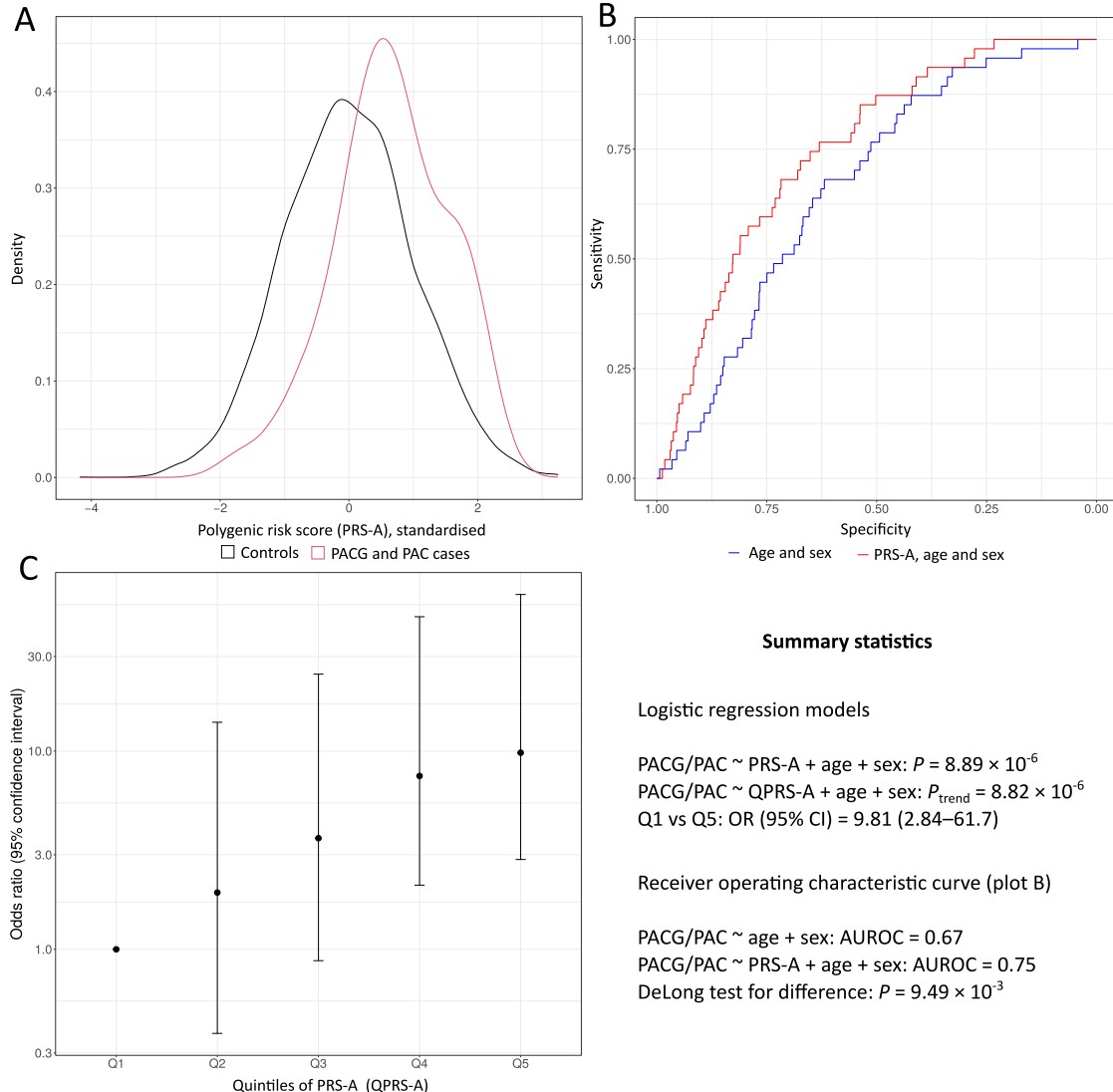

**Summary statistics**

Logistic regression models

PACG/PAC ~ PRS-A + age + sex: $P = 8.89 \times 10^{-6}$
PACG/PAC ~ QPRS-A + age + sex: $P_{trend} = 8.82 \times 10^{-6}$
Q1 vs Q5: OR (95% CI) = 9.81 (2.84–61.7)

Receiver operating characteristic curve (plot B)

PACG/PAC ~ age + sex: AUROC = 0.67
PACG/PAC ~ PRS-A + age + sex: AUROC = 0.75
DeLong test for difference: $P = 9.49 \times 10^{-3}$

**Fig. 4 | Polygenic risk score PRS-A from the MTAG analysis applied to 47 PACG and PAC cases and 6623 controls in the EPIC-Norfolk study. A** Density plots of standardised polygenic risk score A (PRS-A) for PACG and primary angle closure (PAC) cases (red) and controls (black). **B** Receiver operating characteristic (ROC) curves for models of PRS-A, age and sex (red) and of age and sex (blue) predicting PACG case status. Area under the curve (AUC) was calculated for each, and De Long's test was used to compare AUCs. **C** Odds ratio of PACG and PAC cases versus controls by quintiles of multi-trait analysis of genome-wide association study (MTAG) PRS-A adjusted for age and sex with quintile 1 as reference. Data are presented as odds ratios and error bars represent 95% confidence intervals. PACG and PAC are defined clinically. Controls exclude other forms of glaucoma and suspected glaucoma. QPRS-A quintiles of PRS-A.

sex alone had AUROC = 0.67, (difference in AUROC between models (ΔAUROC) 0.08, $P_{DeLong}$ $9.49 \times 10^{-3}$). A monotonic increasing association with PACG/PAC was apparent across quintiles of standardised PRS-A ($P_{trend} = 8.82 \times 10^{-6}$) and quintile 1 (Q1, reference) was significantly different from quintile 5 (Q5), (OR 9.81 (95% CI 2.84–61.7)). Models additionally including spherical equivalent (SE) gave similar associations. A model including PRS-A, SE, age and sex had AUROC = 0.75, while a model including SE, age and sex had AUROC = 0.69 (ΔAUROC 0.06, $P_{DeLong}$ $2.32 \times 10^{-2}$). The association of PACG across quintiles of PRS-A ($P_{trend} = 9.89 \times 10^{-5}$) and Q1 versus Q5 of standardised PRS-A (OR 7.80 (95% CI 2.18–49.90)) for these models adjusted for SE, age and sex (Supplementary Fig. 10) were similar to equivalent models adjusted for age and sex only (Fig. 4). A more hypermetropic SE was significantly associated with PACG in a logistic regression model adjusted for age and sex (OR per dioptre 1.24, 95% CI 1.07–1.44; $P = 4.77 \times 10^{-3}$) (Supplementary Table 7). Further adding PRS-A to this model substantially improved the pseudo $R^2$ (from 3.8 to 5.9%); PRS-A,

after standardisation to mean = 0 and standard deviation = 1, was significantly associated with PACG in this model (OR 1.79, 95% CI 1.32–2.42; $P = 1.83 \times 10^{-4}$) but refractive error was no longer significant (OR 1.13, 95% CI 0.97–1.32; $P = 0.12$).

A PRS (PRS-B) was then generated using the summary statistics from the European ancestry meta-analysis and tested using PACG-related traits in an independent cohort, the EPIC-Norfolk Eye Study[11]. Supplementary Table 8 shows the continuous association and trend over quintiles of the PRS with PACG-related traits. Spherical equivalent (dioptre, positive direction, $P = 3.98 \times 10^{-5}$, adjusted $R^2$ 0.03), axial length (mm, inverse direction, $P = 6.05 \times 10^{-13}$, adjusted $R^2$ 0.06), anterior chamber depth (mm, inverse direction, $P = 9.21 \times 10^{-15}$, adjusted $R^2$ 0.01) and blue eye colour (inverse direction), $P = 2.17 \times 10^{-11}$, adjusted $R^2$ 0.01) were all strongly associated with PRS and directionally consistent with expectations from clinical observation and known PACG risk factors. The results for IOP (mmHg, positive direction, $P = 8.22 \times 10^{-6}$, adjusted $R^2$ 0.02) are in the opposite direction to our

genetic correlation analyses, potentially reflecting greater weighting of PACG-risk variants which are more likely to result in higher IOP. To determine whether our PRS is reflecting risk for angle-closure glaucoma specifically, or glaucoma more generally, we also examined for its predictive ability for POAG in the EPIC-Norfolk Eye Study; the PRS did not discriminate between POAG and controls (positive direction, $P = 0.86$, adjusted $R^2$ 0.002, Supplementary Table 8). The association of PACG-related traits by quintiles of PRS-B is shown graphically in Supplementary Fig. 11. PACG/PAC associations for models using PRS-B were weaker than those using PRS-A. An AUROC = 0.71 for a model including PRS-B, SE, age and sex was non-significantly greater than AUROC = 0.69 for a model for SE, age and sex ($\Delta$AUROC 0.02, $P_{DeLong}$ 0.26). The top quintile of standardised PRS-B was significantly associated with PACG compared to the bottom quintile, adjusted for SE, age and sex, OR 2.26 (95% CI 1.01–5.54). The trend across quintiles was also significant ($P_{trend} = 2.54 \times 10^{-2}$). Similarly, an AUROC = 0.69 for a model including PRS-B, age and sex was not significantly different to an AUROC = 0.67 for a model with just age and sex ($\Delta$AUROC 0.02, $P_{DeLong}$ 0.43) while the association of PACG across quintiles of PRS-B was $P_{trend} = 0.023$ and the association for Q1 versus Q5 of standardised PRS-B was OR 2.41 (95% CI 1.08–5.90) (Supplementary Fig. 12).

It is common for PACG to remain undiagnosed in its earlier phases when the disease is usually asymptomatic. Late diagnosis and delay in treatment significantly increase the risk of blindness[40,41]. While general population screening for PACG is unlikely to be cost-effective, given its relatively low prevalence, our results suggest that targeting screening to people at high genetic risk may be more likely to be successful, though further studies are needed to examine this. Likewise, evidence for the use of PRS to predict PACG severity is limited[42]. While easy-to-measure phenotypes such as eye colour and refractive error could be used to improve risk models, the associations reported here suggest that these would be insufficiently powerful to improve classification. However, the PRS we constructed was able to discriminate between the PACG cases and controls in an independent population. The same risk score showed no difference between cases and controls when applied to POAG.

We also tested the performance of our PRS in participants of non-European ancestry, 137 PACG and acute primary angle closure (APAC) cases and 245 controls, that were excluded during the selection of our European cohort. Using a model that included PRS-A and sex gave an AUROC of 0.63, significantly greater than the AUROC for sex alone of 0.57 ($\Delta$AUROC 0.06, $P_{DeLong}$ 1.66 × 10$^{-2}$), and there was an association with PACG/APAC over quintiles of PRS-A ($P_{trend} = 1.07 \times 10^{-3}$) (Supplementary Fig. 13). We did not include age in these analyses as this was not available for these cohorts. Our results support the predictive utility of the PRS in participants of non-European ancestry, though we acknowledge that further work is needed in larger and more diverse studies before this approach can be implemented clinically.

## Strengths and limitations

Our results provide further evidence of the explainable heritability of PACG. While the association between refractive error and PACG has previously been clinically established, here we show that a genetic predisposition to a smaller eye confers risk for PACG. Importantly, we demonstrate a significant association of our PRS with PACG even when including refractive error. We also demonstrate genetic factors related to iris morphology as increasing risk for PACG, although previous studies have reported only modest improvements in PACG detection of genetic factors over anterior segment imaging parameters[43]. While our current GWAS are underpowered to derive an accurate PRS, we hypothesise that larger discovery sample sizes in the future together with leveraging genetic correlation with refractive error, will enable the development of clinically useful PRS for risk of PACG. Our study has many strengths, but we also acknowledge the limitations adherent to its design. PACG is relatively uncommon in Europeans and despite

our discovery cohort having >500,000 participants, case numbers were modest. Our choice of MAF < 0.01 was arbitrary but attempted to balance minimising false positives without eliminating true positives. While false positives such as PCDH7 were not unexpected, the use of several replication cohorts helped identify them. Reference panels and imputation software differed between the cohorts used in our study. However, we analysed each cohort separately before meta-analysing the results. Importantly, the cases and controls from each study used the same imputation panel, thus minimising any potential biases between cases and controls (which will drive spurious results). PACG was ascertained in our discovery cohort using routine hospital medical coding. While we acknowledge that this may result in a higher degree of misclassification than a direct clinical examination as part of the study, recent studies[44,45] have reported sufficient accuracy for epidemiologic research for PACG from ICD coding. In addition, six of our replication cohorts used multiple clinical criteria for ascertainment and were not reliant on ICD codes from routine healthcare record linkage. While associations in replication cohorts were driven by the larger cohorts, this is a function of sample size rather than any inherent weakness in other cohorts. Although traits such as blue eye colour differ by ancestral group, our PRS was able to demonstrate effectiveness in participants from European and Asian ancestries. While our results suggest potential clinical utility for our PRS over and above a measurement of SE, our validation samples are small, and our simulations suggest that larger GWAS in the future will generate better performing PRSs[46] (Supplementary Fig. 14).

In this large-scale discovery GWAS for PACG in Europeans, we identified four novel loci confirmed with independent replication. While most loci also confer PACG risk in Asian populations, two loci related to eye colour did not, potentially reflecting the more homogeneous eye colour in Asian populations. By combining our European and Asian results, we identified a further six novel loci, many of which relate to axial length and refractive error. We demonstrate that hypermetropia and shorter axial length cause PACG and identify a further 117 novel PACG loci by conducting a multi-trait analysis of PACG with refractive error. PRS derived from our results predicts angle-closure disease in an independent population more strongly than refractive error, opening up the possibility for targeted screening efforts for this blinding disease in the future.

## Methods
### Ethics

The UKB study was conducted with the approval of the North-West Research Ethics Committee (ref 06/MRE08/65), in accordance with the principles of the Declaration of Helsinki, and all participants gave written informed consent. The EPIC-Norfolk Eye Study was approved by the Norwich Local Research Ethics Committee (05/Q0101/191) and East Norfolk & Waveney National Health Service (NHS) Research Governance Committee (2005EC07L), in accordance with the principles of the Declaration of Helsinki and the Research Governance Framework for Health and Social and all participants gave written, informed consent. Italian cohort: The Comitato Etico Interaziendale A.O.U. San Giovanni Battista di Torino approved the study. USA cohort: The study was approved by the IRB of the New York Eye and Ear Infirmary of Mount Sinai, New York, NY. All participants from Iowa provided informed consent, and the study was approved by the University of Iowa's IRB Board. Australian cohort: Ethical approval was obtained from the human research ethics committees of the Southern Adelaide Health Service/Flinders University, and the study was conducted in accordance with the Declaration of Helsinki and its subsequent revisions. Informed written consent was obtained from each individual. Brazilian cohort: Ethical approval was granted by the University of Campinas Research Ethics Committee, Campinas, Sao Paolo, Brazil as Certificate of Presentation for Ethical Appreciation (Certificado de Apresentação para Apreciação Ética or CAAE):

76347317.0.0000.5404. United Kingdom cohort: The study was approved by the Nottingham Research and Ethics Committee and the East Central London Research and Ethics Committee.

## Discovery cohort

Our initial GWAS used European participants of the UKB study. The cohort has been described elsewhere[47,48], but in brief, the UKB is a large population-based cohort study of half-a-million participants recruited between 2006 and 2010 in 22 centres across England, Scotland and Wales. All UK residents aged 40–69 years, registered with a National Health Service (NHS) general practice, and living within 25 miles of a study centre were invited to attend. Participants completed a comprehensive baseline touchscreen questionnaire and were asked to provide a blood sample for genotyping. PACG status was ascertained in the UKB cohort by individual NHS number linkage of UKB to NHS hospital records. Standardised hospital discharge coding using International Classification of Disease (ICD) is routinely collated in all NHS hospitals and held in national databases. We defined PACG as ICD release 10 code H40.2 and ICD release 9 code 365.2. While some PACG might have been misclassified as open-angle glaucoma or other forms of the disease, no differences were observed in the loci identified when the main discovery GWAS was compared with a sensitivity analysis in which all controls with non-PACG glaucoma, suspected glaucoma, and glaucoma-related medication were excluded.

## Replication cohorts

Participants with European ancestry from six independent cohorts were used to replicate the discovery findings (1619 cases, 334,031 controls). PACG and APAC were clinically ascertained in replication cohorts from Italy, UK, USA, Brazil and Australia (557 cases, 7597 controls) while the FinnGen cohort used medical coding from primary and secondary care (1062 cases, 326,434 controls). Consistent with the previous largest GWAS for PACG[8,9], we have included participants with APAC as they are very high risk for developing vision impairment from PACG if left untreated[49]. The combined cohorts had 1619 PACG cases and 334,031 controls, including GWAS summary statistics for 327,496 participants of FinnGen[50]. Details of case definition, inclusion criteria and cohort-specific details of ascertainment are given in Supplementary Note 1 and Supplementary Note 2.

## Identification of European ancestry

Although participants used in the study were from predominantly European countries, we determined the genetic ancestry in each cohort and excluded non-Europeans to reduce confounding from population stratification. Europeans were identified by combining study data with multi-ancestry 1000 Genomes Study data and examining clustering by plotting principal components (PC1 vs PC2 and PC1 vs PC3) as shown in Supplementary Fig. 15. The 27 known genetic ancestries identified in 1000 Genomes were first grouped in 6 colour-coded categories and cluster regions were defined using rectangular areas. The majority of study participants clustered in the same region as Europeans from the 1000 Genomes, while 382 participants outside either region in the two plots were excluded. Participant numbers by cohort for European and non-European participants are shown in Fig. 1.

## Imputation

Genetic imputation was used to increase the number of variants and resolution for directly called genetic data. Imputation of the UKB cohort has been previously described[51] but in brief involved imputation against the UK10K haplotype reference panel merged with the 1000 Genomes Phase 3 reference panel using a modified version of the IMPUTE2 software[51]. In FinnGen, imputation used the population-specific Sequencing Initiative Suomi (SISu) v.3 imputation reference panel and Beagle 4.1[50]. The other replication cohorts required imputation using their directly called data. Pre-imputation quality control

and exclusion were applied to genetic data from each replication cohort using the McCarthy Group Tools (https://www.well.ox.ac.uk/wrayner/tools/). Data were transferred to the Michigan Imputation Server, where imputation was performed using Minimac4 against the Haplotype Reference Consortium (HRC) reference panel consisting of 64,940 haplotypes of predominantly European ancestry. Data were also transferred to the Trans-Omics for Precision Medicine (TOPMED) server[52]. TOPMED uses Minimac4 with the TOPMED reference panel which consists of a larger sample size than HRC with more diverse ancestral backgrounds. We compared the imputation from the two panels and found that GWAS associations were similar but fewer TOPMED variants in replication cohorts were common with UKB or FinnGen. Hence, we have reported associations using HRC-based results throughout. Non-European participants were separately imputed using the 1000 Genomes phase 3, v.5 panel, consisting of genomes of 2504 individuals from 26 populations on the Michigan Imputation Server.

## Genome-wide association study (GWAS)

Quality control (QC) was performed on UKB directly called genotypes with exclusions made for both variants and subjects. Individuals were excluded if their missing call rates exceeded 0.1. Variants with Hardy–Weinberg equilibrium exact test having a $P$ value $\leq 10^{-15}$, a lower minor allele count bound $\leq 100$ or a minor allele frequency (MAF) $\leq 0.01$ were also excluded. Information quality score, a measure of genotyping quality, was assessed and variants $\leq 0.8$ were excluded. A further 178 participants were excluded having opted out of the study. Total participants prior to QC were 442,532 and after QC were 440,749 (202,164 men and 238,585 women with mean age 56.8 years and age range 39–73 years) while 11,234,784 variants remained after QC. Men and women were combined for all analyses as indicated in our study design. Quantile-quantile (QQ) plots were used to examine the extent to which the observed distribution deviated from the null and test for systematic bias. In Supplementary Fig. 16, QQ plots, and lambda statistics are shown for the initial discovery GWAS and European meta-analysis. LD Score regression was performed on the discovery cohort to test for genetic inflation, which can cause spurious associations in GWAS analyses, but this was not evident ($\Lambda$ GC 1.056).

The UKB and each replication cohort were examined for associations with PACG using logistic regression and linear mixed models adjusting for age, sex and the first ten genomic principal components (PCs), using the software Regenie[53]. Additive modelling was adopted a priori to ensure a consistent analytical approach and reduce the likelihood of overfitting. Per-allele genome-wide associations were examined, and dominant and recessive models are shown in Supplementary Table 9. It is possible that a recessive model better fits the association with PACG at rs10818834, given the larger effect size and smaller $P$ value compared to the additive model. UKB and EPIC-Norfolk participants were genotyped using the UKB Axiom Array[51]; genotyping for FinnGen participants used a custom Axiom FinnGen1 array[50]. Genetic data for all cohorts use human genome assembly GRCh37, except for FinnGen which uses GRCh38. Liftover[54] was used to assign GRCh37 chromosome positions for FinnGen.

Independent loci were defined as having a distance greater than 10 Mb, and lead variants were defined as the variant with the smallest $P$ value within a locus. An unbalanced case–control ratio is common in large-scale biobank cohorts[55], and effective sample size can be calculated and used in place of actual sample size. Regenie uses linear mixed models and whole-genome regression models to account for population structure and relatedness. Associations in Regenie were tested using logistic regression. Regenie uses a Firth logistic regression test for unbalanced case–control ratios, to help remove bias from the maximum-likelihood estimates in logistic models[53]. Effective sample size (ESS) was used in the arguments provided for COJO[56], METAL[57] and MTAG[37] which were developed based on the balanced case–control

assumptions[58]. ESS was calculated using the formula:

$$EffectiveN = \frac{2}{\frac{1}{NCases} + \frac{1}{NControls}}$$

## Meta-analyses

As outlined in Fig. 1, meta-analyses were performed on the six European replication cohorts shown graphically in Supplementary Fig. 3; in the 14 Asian cohorts; in all Europeans using UKB discovery and the six European replication cohorts as shown in Fig. 2; and lastly in Europeans and Asians using all 21 cohorts (Supplementary Fig. 5). All meta-analyses were performed using METAL[57] with fixed effects, weighting the effect size estimates using the inverse of the corresponding standard errors (inverse variance-based approach). Quality control parameters were applied, with genotypes excluded where imputation information scores were <0.4 and for MAF < 0.005. ESS was used in place of total N for controls.

## Conditional analysis

Conditional and joint multiple-SNP analysis of the summary statistics was performed using GCTA COJO[56] to identify independent effects within each locus and to calculate the variance explained associated with the trait after the conditional analyses. COJO performs a stepwise variant selection procedure based on conditional $P$ values and estimates the joint effects of all selected SNPs after the model optimisation. A genomic region based on a 10 Mb window centred on the locus was used, the threshold of significance was set at $P < 5 \times 10^{-8}$, and the collinearity threshold was set at $r^2 = 0.9$. The LD estimates were derived from a random subset of 10,000 UKB participants chosen as the reference sample since the largest participating cohort is recommended by the software authors for this purpose.

## Genetic correlation analysis

To assess shared genetic architecture between our meta-analysis results and PACG-related traits, genetic correlations were calculated using linkage disequilibrium (LD) score regression[36]. GWAS Summary statistics were grouped into five categories: anthropometry (standing height and waist circumference), education (educational attainment), glaucoma-related IOP, vertical cup to disc ratio (VCDR), macular retinal nerve fibre layer (RNFL) thickness and ganglion cell–inner plexiform layer (GCIPL) thickness, pigmentation (darker skin colour and black hair colour) and refraction (axial length, refractive error, hypermetropia, age when started wearing glasses and cataracts). Quality control and standardisation were applied to each summary statistic prior to determining their genetic correlations. A heatmap plot using colour shades to represent positive and negative correlation was constructed using the R package ggcorrplot (Supplementary Fig. 6).

## Mendelian randomisation (MR)

To examine potential causal relationships, two-sample MR analysis was performed for each of the exposures (axial length, eye colour and refractive error) on PACG. Independent variants associated with the exposure at genome-wide significance were selected as instrumental variables (IV), with variant-outcome associations assessed in the PACG European ancestry meta-analysis. After allele harmonisation and removal of palindromic variants (MAF > 0.42), a multiplicative random-effects inverse-variance weighted (IVW)[59] approach was utilised for the main analysis. Sensitivity analyses were performed using four robust methods: weighted median[60], weighted mode[61], MR-Egger[62] and MR-PRESSO[63]. Axial length used all genome-wide significant variants from a GWAS using the GERA cohort[64]. After LD clumping (10,000 kb; $r^2$ 0.001), 13 independent variants were selected as IVs. For eye colour, a GWAS using 23andMe and the VisiGen Consortium was used[13]. This is the largest GWAS of eye colour available; the phenotype was not measured in UK Biobank. Using the reported independent genome-wide significant variants, 41 SNPs were utilised as IVs. For refractive error, all genome-wide significant variants from a GWAS using UK Biobank and GERA[29,65] were used with 377 independent variants selected as IVs after LD clumping. We use the largest GWAS for refractive error[29] in order to use the most accurate information to examine causality. The study reported findings using Z-scores rather than dioptres since this study in part used refractive error inferred from questionnaire data. Therefore, the final results from our MR analysis re on a Z-score scale rather than a dioptre scale. The analyses used R packages MendelianRandomization[66], MR-PRESSO[63] and TwoSampleMR[67,68].

## Multi-trait analysis for GWAS

In order to increase power to discover novel associations with PACG, Multi-trait Analysis for GWAS (MTAG) was used by combining the PACG European ancestry meta-analysis with refractive error. The MTAG analysis used the method described by Turley[37] and associated software. Summary statistics from the European ancestry meta-analysis ($N$ effective = 18,885) were combined with summary statistics for refractive error[29]. To enable us to best identify independent loci among the multiple significant genomic regions, we carried out a conditional analysis.

## Polygenic risk score

We created two PRSs. Summary statistics from our MTAG analysis were used to create a PRS (PRS-A) to examine the PACG association in an independent cohort, the EPIC-Norfolk Eye Study[69]. A second PRS (PRS-B) was created using summary statistics from our European ancestry meta-analysis which allowed us to examine PACG-related traits in EPIC-Norfolk without the PRS score being influenced by refractive error, which was used in the construction of the MTAG. The PRS scores were derived using SBayesRC[39], a Bayesian method that makes use of functional consequence annotation data that can influence the probability of a variant being considered causal, as well as the magnitude of its causal effect size. We utilised the provided genomic annotation data and LD reference for Europeans from HapMap3.

Associations between the PACG PRSs and PACG status in the EPIC-Norfolk Eye Study were tested[11]. PACG, which included PAC cases, was ascertained by clinical examination (including gonioscopy) by a glaucoma specialist in the EPIC-Norfolk Eye Study (47 cases, 6623 controls)[11,70]. Controls excluded participants with other forms of glaucoma or glaucoma suspects. Prediction accuracy was estimated by calculating the AUROC for PACG status. Comparison between two AUCs was performed using the non-parametric DeLong statistical test[71]. The European ancestry meta-analysis-derived PRS examined associations over PRS quintiles and axial length (AL, mm), anterior chamber depth (ACD, mm), spherical equivalent (SE, dioptre), blue eye colour, POAG and intraocular pressure (IOP, mmHg). Logistic regression was used to test for the association between quintile 1 (reference) and other quintiles and trend was tested across quintiles. Continuous quantitative traits (AL, ACD, SE and IOP) were additionally modelled after being standardised with mean 0 and standard deviation 1. All models were adjusted for sex and age at the time of the EPIC-Norfolk Eye Study. Genotyping in EPIC-Norfolk used the Affymetrix UKB Axiom Array with exclusions made after variants and sample quality control. Imputation was performed by the Sanger Imputation Service using the IMPUTE4 software and Haplotype Reference Consortium and UK10K plus 1000 Genomes phase 3 reference panels. Variants with a genotyping quality score (INFO) < 0.4 were excluded. The choice of imputation score cut-off is arbitrary and unlikely to have a large impact on our results, given that our genome-wide significant loci all had an imputation score >0.85 (Table 1).

## Software

Our analyses used publicly available software. GWAS was performed using Regenie (v3.2.7). Regenie was chosen for our analyses pipeline because of its strong capability for handling cryptic relatedness using Firth regression which is suitably implemented and given the relatedness structure in UK Biobank. Conditional and joint analysis of the summary statistics used GCTA COJO (v1.94.1). Quality control filtering of VCF files was performed using BCFtools (v1.9) and PLINK (v1.90b). Polygenic risk scores were calculated using SBayesRC (v 0.2.3). Meta-analysis was performed using METAL (v 2020-05-05). METAL is a well-established meta-analysis software system. We aimed to identify homogenous associations and therefore used a fixed-effects meta-analysis package. Fixed effects analyses are more likely to identify loci with more consistent effects across ethnic groups. LD Score regression used LDSC (v 1.0.1). Regional association analyses used LocusZoom (v 1.4). Variants were annotated using Ensembl Variant Effect Predictor v101 with assembly GRCh38. The code for the software is available on the website of each package. Data manipulation and analysis were performed using R-4.3 with packages dplyr, gtools, tibble and tidyr. Manhattan plots were created using the R packages topr and ggplot2. MR was performed using the R packages MendelianRandomization (v 0.10.0), MR-PRESSO (v 1.0) and TwoSampleMR (v 0.5.10). Simulations of GWAS sample size estimates (Supplementary Fig. 14) used GENESIS (v 1.0). Proxy variants were obtained using the R package LDlinkR (v 1.4.0). The code for each of the R packages can be found in their associated vignettes.

## Reporting summary

Further information on research design is available in the Nature Portfolio Reporting Summary linked to this article.

# Data availability

We obtained data files containing directly called and imputed genetic data from UK Biobank and EPIC-Norfolk. Cohorts from Australia, USA, UK, Italy and Brazil provided directly called genetic data. A UK Biobank dataset containing hospital ICD10 codes was used to derive PACG and contained age and sex. A dataset from the EPIC-Norfolk Eye Study contained PACG outcome, age and sex. A dataset of GWAS summary statistics for PACG was obtained from FinnGen. Access to UK Biobank data requires an application in order to protect the privacy of participants and to conform to confidentiality and data governance policies. Requests for access to UK Biobank data should be made to the UK Biobank Access Management Team (access@ukbiobank.ac.uk). Access to EPIC-Norfolk data requires an application in order to protect the privacy of participants and to conform to confidentiality and data governance policies. Requests for access to EPIC-Norfolk data can be via https://www.epic-norfolk.org.uk/for-researchers/data-sharing/data-requests. Access to FinnGen summary statistics can be made via https://www.finngen.fi/en/access_results. Contact details of national replication cohorts can be found in Supplementary Note 1 and Supplementary Note 2. The summary-level data generated in this study are provided in the Source Data file. Source data are provided with this paper.

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

## Acknowledgements

This research was supported by the NIHR Biomedical Research Centre at Moorfields Eye Hospital and the UCL Institute of Ophthalmology. This research has been conducted using the UK Biobank Resource under Application Number 36741. UK Biobank is funded by the Medical Research Council, Department of Health, Wellcome Trust medical charity, Scottish Government and Northwest Regional Development Agency. The UKB study protocol and detailed description of measurements are available from (https://www.ukbiobank.ac.uk). The study was conducted with the approval of the North-West Research Ethics Committee (ref 06/MRE08/65), in accordance with the principles of the Declaration of Helsinki, and all participants gave written informed consent. The EPIC-Norfolk study (https://doi.org/10.22025/2019.10.105.00004) has received funding from the Medical Research Council (MR/N003284/1 and MC-UU_12015/1) and Cancer Research UK (C864/A14136). The genetics work in the EPIC-Norfolk study was funded by the Medical Research Council (MC_PC_13048). We are grateful to all the participants who have been part of the project and to the many members of the study teams at the University of Cambridge who have enabled this research. We would like to acknowledge the participants and investigators of all the cohorts used in this study, including UKB, EPIC-Norfolk, replication cohorts FinnGen and those from UK, USA, Brazil, Australia and Italy. A.P.K. is supported by a UK Research and Innovation Future Leaders Fellowship, an Alcon Research Institute Young Investigator Award, and a Lister Institute for Preventive Medicine Award. This research was supported by the NIHR Biomedical Research Centre at Moorfields Eye Hospital and the UCL Institute of Ophthalmology. A.P.K., P.J.F., R.N.L., M.I.B., K.V.S., R.H. and Z.S. were supported by a grant from the National Institute for Health and Social Care Research (NIHR) for a Biomedical Research Centre (BRC) at Moorfields Eye Hospital NHS Foundation Trust and UCL (University College London) Institute of Ophthalmology. A.P.K. was supported by grants from UK Research and Innovation (UKRI) Future Leaders Fellowship [MR/Y033930/1], Alcon Research Institute Young Investigator Award, and Lister Institute Fellowship. P.J.F. and K.V.S. are supported by grants from Fight for Sight (UK) and The Desmond Foundation. P.J.F. was supported by an unrestricted grant from The Alcon Research Institute. The funding organisations had no role in the design or conduct of this research.

## Author contributions

A.P.K., P.J.F., CC.K. and T.A. conceived the study. A.P.K. supervised this work. P.H. and A.P.K. directed the overall analysis. R.N.L. performed the quality control, imputation, GWAS, replication analyses, meta-analyses, conditional analyses and genetic correlation analyses. R.N.L. developed the bioinformatics and computational pipelines. K.V.S. performed the MR analysis and created MR figures. M.B. performed the PRS and MTAG analyses. R.N.L. drafted the manuscript, tables and figures. All authors (R.N.L., M.I.B., K.V.S., R.H., Z.S., Z.L., N.W., T.D., C.P.P., M.N., K.T., Y.I., Y.T., M.T., N.O., M.U., C.S., S.K., K.M., N.K., A.F., M.B.M., J.P.C.V., V.P.C., M.E.N., C.L.H., S.A.P., J.E.C., A.K., AT.L-S., E.U.L., K.H.P., L.V., R.G., T.A., CC.K., P.J.F., P.H. and A.P.K.) reviewed and approved the paper.

## Competing interests

A.P.K. has acted as a paid consultant or lecturer to AbbVie, Aerie, Allergan, Google Health, Heidelberg Engineering, Novartis, Reichert, Santen, Thea and Topcon. P.J.F. has acted as a paid consultant to AbbVie, Alphasights, GLG, Google Health, Guidepoint, PwC and Santen outside the submitted work. The remaining authors declare no competing interests.

## Additional information

**Robert N. Luben** ®[1,2,25] ✉, **Mahantesh I. Biradar** ®[1,25], **Kelsey V. Stuart** ®[1,25], **Ruiqi Hu** ®[1], **Zihan Sun**[1], **Zheng Li** ®[3], **Ningli Wang** ®[4], **Tan Do**[5,6], **Chi Pui Pang**[7], **Masakazu Nakano** ®[8], **Kei Tashiro**[8], **Yoko Ikeda**[9], **Yuichi Tokuda** ®[8], **Masami Tanaka**[8], **Natsue Omi**[8], **Morio Ueno** ®[9], **Chie Sotozono**[9], **Shigeru Kinoshita** ®[10], **Kazuhiko Mori** ®[9], **Naris Kitnarong**[11], **Antonio Fea** ®[12], **Mônica B. Melo** ®[13], **José Paulo C. Vasconcellos**[14], **Vital P. Costa**[14], **Monisha E. Nongpiur**[15,16], **Ching Lin Ho**[15,16], **Shamira A. Perera**[15,16], **Jamie E. Craig** ®[17], **Antonia Kolovos** ®[17], **Ahmad Tajudin Liza-Sharmini**[18], **Edgar U. Leuenberger**[19], **Ki Ho Park** ®[20], **Lingam Vijaya**[21], **Ronnie George**[21], **Tin Aung**[15,16,25], **C. C. Khor** ®[3,15,16,25], **Paul J. Foster** ®[1,25], **Pirro Hysi** ®[22,23,24,25] & **Anthony P. Khawaja** ®[1,2,25]

[1]NIHR Biomedical Research Centre, Moorfields Eye Hospital NHS Foundation Trust and UCL Institute of Ophthalmology, London, UK. [2]MRC Epidemiology Unit, School of Clinical Medicine, University of Cambridge, Cambridge, UK. [3]Genome Institute of Singapore, Agency for Science, Technology and Research, Singapore, Singapore. [4]Beijing Tongren Eye Center, Beijing Tongren Hospital, Capital Medical University, Beijing Ophthalmology & Visual Science Key Lab,

Beijing, China. [5]Vietnam National Eye Hospital, Hanoi, Vietnam. [6]University of Medicine and Pharmacy, Vietnam National University, Hanoi, Vietnam. [7]Department of Ophthalmology and Visual Sciences, The Chinese University of Hong Kong, Shatin, N.T, Hong Kong. [8]Department of Genomic Medical Sciences, Kyoto Prefectural University of Medicine, Kyoto, Japan. [9]Department of Ophthalmology, Kyoto Prefectural University of Medicine, Kyoto, Japan. [10]Frontier Medical Science and Technology for Ophthalmology, Kyoto Prefectural University of Medicine, Kyoto, Japan. [11]Department of Ophthalmology, Faculty of Medicine Siriraj Hospital, Mahidol University, Bangkok, Thailand. [12]Dipartimento di Scienze Chirurgiche, Università di Torino, Turin, Italy. [13]Centro de Biologia Molecular e Engenharia Genética, Universidade Estadual de Campinas, Campinas, Brazil. [14]Departamento de Oftalmologia, Faculdade de Ciências Médicas, Universidade Estadual de Campinas, Campinas, Brazil. [15]Duke-NUS Medical School, Singapore, Singapore. [16]Singapore Eye Research Institute, Singapore National Eye Centre, Singapore 168751, Singapore. [17]Department of Ophthalmology, Flinders Health and Medical Research Institute, Bedford Park, SA, Australia. [18]Department of Ophthalmology and Visual Science and Hospital Pakar Universiti Sains Malaysia, Universiti Sains Malaysia, Kota Bharu, Kelantan, Malaysia. [19]Asian Eye Institute and University of the East Ramon Magsaysay College of Medicine, Manila, Philippines. [20]Department of Ophthalmology, Seoul National University Hospital, Seoul, Republic of Korea. [21]Sankara Nethralaya, Medical Research Foundation, Chennai, Tamil Nadu, India. [22]Twin Research and Genetic Epidemiology and Ophthalmology, King's College London, London, UK. [23]Institute of Child Health, University College London, London, UK. [24]Sørlandet Sykehus Arendal, Arendal, Norway. [25]These authors contributed equally: Robert N. Luben, Mahantesh I. Biradar, Kelsey V. Stuart, Tin Aung, C. C. Khor, Paul J. Foster, Pirro Hysi, Anthony P. Khawaja. ✉e-mail: r.luben@ucl.ac.uk

