## [Transparent Peer Review file · Nature Communications]

GWAS for primary angle-closure glaucoma identifies novel loci related to ocular biometry and morphology

Corresponding Author: Dr Robert Luben

Version 0:

Reviewer comments:

Reviewer #1

(Remarks to the Author)

Luben and colleagues performed GWAS meta-analyses of the trait, 'primary angle-closure glaucoma' (PACG), a common subtype of glaucoma. Prior genetic studies have identified few loci for this trait, and it has not been clear how the genetic contribution to PACG varies across East Asian (EAS) and European (EUR) ancestries. The current work provided a major advance in knowledge. Many novel loci for PACG in EAS and EUR ancestry populations were identified, along with a novel link to eye color.

The study utilized standard, robust methodology for GWAS, meta-analysis, genetic correlation analysis, and Mendelian randomization (MR). Apart from my comment #1, my other comments relate to relatively minor issues where additional clarification would be valuable.

1. Line 241-285. The presentation of the polygenic risk score (PRS) results appeared contrived, as if the authors were seeking to present an overly optimistic assessment. In particular, it was not clear how much prediction accuracy was provided by the PRS once refractive error had been accounted for. I recommend: (a) give the two polygenic risk scores names, such PRS-A and PRS-B, to help the reader differentiate which score is used in which analysis; (b) As well as age and sex, include refractive error in the 'baseline' AUROC analysis for PACG, since refractive is a well-known and easily measured risk factor for PACG; (c) report delta-AUROC values instead of raw AUROC values, where delta-AUROC is the AUROC for the PRS + Age + Sex + refractive error minus the AUROC for the Age + Sex + refractive error:

$$\text{delta-AUROC} = \text{AUROC} [\text{PRS} + \text{Age} + \text{Sex} + \text{refractive error}] - \text{AUROC} [\text{Age} + \text{Sex} + \text{refractive error}]$$

(d) Provide the AUROC result for PACG in the EPIC-Norfolk Eye Study cohort first (line 247), before reporting the results for other traits; (e) At the conclusion of this section and/or in the conclusion of the manuscript, explain (i) how much better the PRS would need to be for use clinically, (ii) the potential sub-optimal performance of the PRS in non-EUR ancestries due to trans-ancestry portability, and the ethical impact this may have on clinical implementation; (iii) estimate how many more PACG cases will need to be recruited to increase the accuracy of the PRS sufficiently for clinical use.

2. GWAS in UK Biobank. (a) The methods section reported that only 509,485 were studied in the UK Biobank GWAS for PACG. Was this a typo, or were only directly-genotyped variants studied? If the latter, why use directly-genotyped variants for UK Biobank but imputed variants for the other cohorts? (b) One of the genome-wide significant variants (PCDH7) had a MAF=0.01 and did not replicate. Such false positives are common in a GWAS with a small number of cases, as was true here (n=1564 cases). What was the statistical power to detect (e.g. an OR>2 or an OR<0.5) a genome-wide significant variant for MAFs <0.05? The authors may wish to choose a more stringent MAF threshold for the discovery GWAS, to reduce the risk of false positives.

3. Mendelian randomization. (a) When testing for a causal effect of refractive error on PACG, it would be more informative to use GWAS summary statistics for refractive error in units of diopters. The current MR effect estimate is expressed as an OR per unit increase in a GWAS Z-score for refractive error, which is not easy to interpret. An MR effect estimate in units of OR per D would be easily understood. (b) According to Supplementary Table 6, the GWAS summary statistics for the trait 'hypermetropia' were obtained from the MRC-IEU database. In my opinion, none of the available hypermetropia summary statistics in the MRC-IEU database are suitable for MR. The summary statistics either have a very small sample size, or they

assumed that (UK Biobank) participants not asked about their reason for wearing glasses were unaffected, i.e. they were classified as controls. For a common trait such as hypermetropia, such massive misclassification will yield incorrect beta coefficients, which will severely bias the MR causal estimate. (c) The exact set of GWAS summary statistics used for each MR analysis should be presented in Supplementary Table 6. For example, currently, the label 'MRC-IEU' for hypermetropia does not make it clear which of the 6 sets of hypermetropia summary statistics available in the MRC-IEU database was used.

4. Supplementary tables and figures. (a) It would be informative to include Supplementary Table 3 in the main text, instead of the supplement. Please add the MAF in each ancestry group to this table. (b) Ticks are used to indicate the level of statistical support in Supplementary Table 4. Why not simply present the exact p-value? (c) Ticks are used to indicate prior reports of association in Supplementary Table 5. Instead, I recommend citing the relevant publication directly. (d) Define the error bars in Supplementary Figure 4.

5. GWAS in UK Biobank and Europeans. The effect sizes of the PACG loci are strikingly high, e.g. OR > 1.2. The authors may wish to examine if these loci show any evidence of dominance or recessive association, or whether an additive model provides a better fit (see, for example, PMID: 32227305).

6. Line 154. The sentence beginning, "Regional association..." Did not link well with the preceding sentences. Could this point be made elsewhere?

Reviewer #2

(Remarks to the Author)

This paper reports the first large-scale genome-wide association study (GWAS) to investigate genetic risk for Primary Angle-Closure Glaucoma (PACG) in Europeans in the UK Biobank (UKB; 1564 cases and 439,185 controls). They detected genome-wide significance at five loci, including three novel loci and two loci previously associated with PACG in Asian populations. They then examined the top associated variants in six independent replication cohorts (1619 cases and 334,031 controls from six national cohorts: Australia, Brazil, Italy, Finland, UK and USA). In meta-analysis of the replication cohorts, four variants were at least nominally significant ($P < 0.05$) and all had consistent direction of effect. To increase power to detect PACG associations, they meta-analysed all European cohorts, finding a total of 10 genome-wide significant loci. They also compared results to previously published findings in Asian-focused analyses and, to further increase power, they performed a multi-ancestry meta-analysis combining their European ancestry meta-analysis with 14 cohorts from Asian countries including China, Vietnam, Thailand, Malaysia, Singapore, Japan and India. An additional 12 novel loci reached genome-wide significance in the multi-ancestry analysis that were not independently significant in the Asian or European ancestry meta-analyses. They performed conditional analyses to identify independent effects within each locus, genetic correlation analysis to assess shared genetic architecture between PACG and PACG-related traits, and Mendelian randomisation (MR) using multiple software packages to examine potential causal relationships between PACG and axial length, eye colour, and refractive error. They performed multi-trait analyses with MTAG and conditional analyses, and generated two polygenic risk scores (PRS) from the MTAG analysis and the European meta-analysis, both of which were tested in the independent EPIC-Norfolk Eye Study to determine association between the PRS and PACG, axial length, anterior chamber depth, spherical equivalent, blue eye colour, primary open-angle glaucoma, and intraocular pressure.

This manuscript is very well written and, for the most part, clearly presented. I have a few questions, suggestions, and concerns, as detailed below, that I think will clarify and possibly improve the manuscript.

-Considering the case definition in UKB, how likely is it that an individual would have an ICD-9 or ICD-10 code for PACG but not truly have one of those diagnoses?

-Given that the main paper methods only list ICD-9 and ICD-10 codes for PACG, the Supplementary Methods' mention of the following in the inclusion criteria was confusing: "Patients with acute primary angle-closure (APAC) or primary angle-closure glaucoma (PACG) were recruited," especially given that the title includes "PACG" and the main paper focuses on PACG. Further, is it typical to jointly analyze PACG with APAC patients? Please provide rationale.

-The Principal Components analysis (Supplementary Figure 12) and Methods describe exclusion of individuals who are not European from the replication cohorts; how likely are these findings to apply to individuals of predominantly non-European ancestry? Would it be possible to evaluate the novel loci and performance of the risk scores in those individuals who were excluded from initial analyses due to more heterogeneous ancestry, or in another ancestral group?

-In Supplementary Figures 10 and 11, Polygenic risk score from the European meta-analysis and spherical equivalent, axial length, anterior chamber depth and blue eye colour in 7,223 EPIC-Norfolk participants (10) and Polygenic risk score from European meta-analysis applied to 47 PACG cases and 6,623 controls in the EPIC-Norfolk study (11) are presented. What are the implications of these findings in Europeans, as well as other ancestral groups?

-The reference panels and imputation software differ between the different groups analyzed; please provide rationale for why this would not potentially influence/increase the likelihood of detecting spurious results (especially when some of the datasets are obviously much larger than others and clearly driving significance of signal detection).

-Please provide justification for the imputation information score cutoff used (< 0.4).

-Please provide a reference for "McCarthy Group Tools."

-Please provide justification for use of Regenie (as opposed to other software, e.g. GENESIS, SAIGE, and fastGWA-GLMM).

-Figure 1 should be updated for clarity and consistency.

-Please explain the use of METAL for the multi-ancestry meta-analysis as opposed to a program developed specifically to take multi-ancestry into account (e.g. MR-MEGA).

-Please elaborate on limitations/weaknesses of the study. E.g. in the replication meta-analysis, the authors acknowledge that the associations were largely driven by the FinnGen, UK, and Australian cohorts, but elaboration on the implications

would be helpful.

-After reading this, I am impressed by the multitude of analyses presented and the novel (and consistently replicated) results. What I am left wondering is - what is the point? I think it would be very helpful to provide additional discussion on the value these results bring to the field.

Reviewer #3

(Remarks to the Author)

This is an excellent and very interesting paper which describes the results of a Genome-Wide Association study for primary angle-closure glaucoma (PACG), and identified novel loci related to ocular biometry and morphology.

Although previous work has been conducted on primarily Asian populations, where PACG is more common, this new study includes two large European cohorts, the UK Biobank and the EPIC-Norfolk studies.

The most significant associations were HERC2/OCA2 (rs12913832:G) previously associated with brown iris colour and a second locus near SEMA3A (rs17245595:C) which has been associated with morphological characteristics and colour of the iris. The morphology previously published was iris crypts but this is not mentioned in the paper.

The authors also identified loci at PXDNL (rs11984688:A), and GLIS3 (rs746970:G), both previously associated with PACG in Asian populations), and at PCDH7 (rs184176302:C)

Six novel loci identified by meta-analysis included three genes previously associated with refractive error: LAMA2 (rs12193446:A), KCNQ5 (rs7744813:C) and DENND1A (rs10818834:T), Mendelian Randomisation confirmed the association of shorter axial length and hyperopia with PACG. Polygenic Risk Score showed association with shallower anterior chamber and higher intraocular pressure.

This is an important advance in the understanding of PACG.

Suggestions

The authors claim this is the FIRST large scale GWAS of European PACG with 1,564 cases, but the previous study (Khor 2016 including some of the current authors), which although predominantly Asian included 1,105 European PACG cases which were used in the replication of the current study. Thus the authors should change the claim of primacy.

The authors should mention that GLIS3 has also been associated with Congenital Glaucoma and with POAG.

The major changes I recommend to the discussion are given below:

I was stunned the authors completely missed all the previous literature linking the phenotypes of HERC2/OCA2 and SEMA3A to PACG, especially when some were involved in these earlier papers.

The previous SEMA3A association was with iris crypt frequency (Larsson 2011).

One of the senior authors from Singapore is on another paper where they correlated iris crypts with Primary Angle Closure Glaucoma.

Koh V, Chua J, Shi Y, Thakku SG, Lee R, Nongpiur ME, Baskaran M, Kumar RS, Perera S, Aung T, Cheng CY. Association of iris crypts with acute primary angle closure. *Br J Ophthalmol*. 2017 Oct;101(10):1318-1322.

The Singaporean group also documented

“Lighter coloured irides with more crypts and/or more furrows were also associated with less convexity.”

Tun TA, Chua J, Shi Y, Sidhartha E, Thakku SG, Shei W, Tan MC, Quah JH, Aung T, Cheng CY. Association of iris surface features with iris parameters assessed by swept-source optical coherence tomography in Asian eyes. *Br J Ophthalmol*. 2016 Dec;100(12):1682-1685.

“irises with more crypts and lighter color were associated with wider angle”

Sidhartha E, Nongpiur ME, Cheung CY, He M, Wong TY, Aung T, Cheng CY. Relationship between iris surface features and angle width in Asian eyes. *Invest Ophthalmol Vis Sci*. 2014 Oct 23;55(12):8144-8.

Thus the authors statement on line 74 of the abstract is not correct and needs to be changed as variation in eye colour can be seen in Asians “Additionally, the effect of variation in iris colour may only be detectable in European populations where colour is more heterogenous.”

“Although few Chinese persons have multiple crypts on their irides, irides with more crypts were significantly thinner and lost more volume on pupil dilation. In view that the latter feature is known to be protective for acute angle-closure”

Chua J, Thakku SG, Tun TA, Nongpiur ME, Tan MC, Girard MJ, Wong TY, Quah JH, Aung T, Cheng CY. Iris Crypts Influence Dynamic Changes of Iris Volume. *Ophthalmology*. 2016 Oct;123(10):2077-84.

The issue of iris crypts and glaucoma needs to be discussed in much more detail.

Whilst iris structural abnormalities are well known in Anterior Segment Dysgenesis and Rieger Syndrome, both of which are associated with open angle glaucoma, there are some other syndromes that can be discussed such as:

Iris ectropion syndrome (Ectropion uveae) is associated with open angle glaucoma

“The posterior pigment epithelium of the iris is found on the anterior stroma at birth. The iris surface is glassy smooth and devoid of crypts.”

Sridhar U, Tripathy K. Iris Ectropion Syndrome. 2023 Aug 25. In: StatPearls [Internet]. Treasure Island (FL): StatPearls Publishing; 2024 Jan–.

“Anterior segment abnormalities included absent iris crypts” is a feature of Knobloch Syndrome (COL18A1)

Hull S, Arno G, Ku CA, Ge Z, Waseem N, Chandra A, Webster AR, Robson AG, Michaelides M, Weleber RG, Davagnanam I, Chen R, Holder GE, Pennesi ME, Moore AT. Molecular and Clinical Findings in Patients With Knobloch Syndrome. *JAMA Ophthalmol*. 2016 Jul 1;134(7):753-62.

In Nanophthalmic eyes, a syndrome closer to PACG where small eyes often develop glaucoma.

“The anterior surface of the iris seemed “smoother” in nanophthalmic eyes than in normal eyes.”

Lu Q, He W, Lu Y, Zhu X. Morphological features of anterior segment: factors influencing intraocular pressure after cataract surgery in nanophthalmos. *Eye Vis (Lond)*. 2020 Sep 9;7:47.

However, the earliest description of the association of iris crypts and narrow angle glaucoma appears to be from 1954.

Posner looking at patients with unilateral narrow angle glaucoma noted the eyes with glaucoma had a smooth stroma, while the eyes without glaucoma had well circumscribed iris crypts.

Posner A. Role of peripheral iris crypts in protection from narrow angle glaucoma. *Eye Ear Nose Throat Mon*. 1954 Jul;33(7):427-8;

And also

Posner A. Peripheral iris crypts in relation to medical and surgical therapy of narrow-angle glaucoma. *Am J Ophthalmol*. 1955 Oct;40(4):469-74.

The authors might consider reviewing the other iris crypt associated genes from the Larsson 2011 paper in their PACG cohort.

Minor comments:

The authors might point out that eye colour was not available from the UK Biobank.

Line 93, “Three quarters of the global disease burden is estimated to occur in Asia”. Suggesting adding “of PACG”.

Lines 262-274 and 280-284: The authors show that adding spherical equivalent refraction (SE) to a classification model including age, sex and PRS slightly improves detection of PACG from AUROC 0.69 to 0.75 but suggest “these would be insufficiently powerful to improve risk models”. Given that SE is easier and cheaper to measure than a PRS, a relevant question may be does age + sex + SE + PRS perform better than age + sex + SE?

Line 294-295: “PRS...predict a substantial proportion of angle-closure disease” It would be better to quantify “substantial” here by providing sensitivity and specificity estimates and comparing to age/sex/SE as relevant.

References 40 and 42 are duplicates

References 11 and 61 are duplicates

The replication cohort included six national cohorts from Australia, Brazil, Finland, Italy, UK and USA.

Whilst the new cohorts from Brazil and Italy include authors from those countries and the FinnGen is a publicly available resource, there are no Australian or US authors on the current paper.

Are those GWAS available from the earlier Khor 2016 paper (1,105 European cases)?

Reviewer #4

(Remarks to the Author)

Version 1:

Reviewer comments:

Reviewer #1

(Remarks to the Author)

The revised manuscript is much improved. All of my original comments have been either addressed or rebutted.

I have one new comment: In Table 1 information was missing for the last 2 columns.

Reviewer #2

(Remarks to the Author)

My concerns have been addressed; this is a fantastic paper.

Reviewer #3

(Remarks to the Author)

The authors have addressed the main issues raised by the reviewers.

Reviewer #4

(Remarks to the Author)

Remarks to the Author		Changes made in the manuscript
Reviewer #1		Lines numbers shown for tracked changes document Additional text is underlined ; deleted text is struck through .
Luben and colleagues performed GWAS meta-analyses of the trait, ‘primary angle-closure glaucoma’ (PACG), a common subtype of glaucoma. Prior genetic studies have identified few loci for this trait, and it has not been clear how the genetic contribution to PACG varies across East Asian (EAS) and European (EUR) ancestries. The current work provided a major advance in knowledge. Many novel loci for PACG in EAS and EUR ancestry populations were identified, along with a novel link to eye color. The study utilized standard, robust methodology for GWAS, meta-analysis, genetic correlation analysis, and Mendelian randomization (MR). Apart from my comment #1, my other comments relate to relatively minor issues where additional clarification would be valuable.	Thank you for your helpful comments. We have made a number of changes to the manuscript based on your suggestions which we believe have improved the clarity and interpretation of our findings.	

Remarks to the Author		Changes made in the manuscript
1. Line 241-285. The presentation of the polygenic risk score (PRS) results appeared contrived, as if the authors were seeking to present an overly optimistic assessment. In particular, it was not clear how much prediction accuracy was provided by the PRS once refractive error had been accounted for. I recommend:	We agree that there should be increased clarity on the contribution of the PRS over and above refractive error (see below).	

(a) give the two polygenic risk scores names, such PRS-A and PRS-B, to help the reader differentiate which score is used in which analysis;

We agree with the reviewer's suggestion and have named the polygenic risk scores (PRS) in the manuscript, figure and tables as PRS-A (constructed from the MTAG analysis) and PRS-B (constructed from the European meta-analysis).

We have made the following changes in the "Population level prediction performance using PRS" section.

Lines: 256-258

We first ~~then~~ created a PRS ("PRS-A") using the MTAG analysis to maximise PACG discovery power and this was tested on PACG ascertained in the independent EPIC-Norfolk Eye Study.

Lines: 258-261

In Figure 4, we examined whether PRS-A ~~the PRS~~ could predict PACG or PAC cases status compared to controls (defined as study participants excluding those with other forms of glaucoma and suspected glaucoma).

Line: 261

The area under the receiver operating characteristic curve for PRS-A, age and sex was 0.75

Line: 280

~~First a~~ A PRS ("PRS-B") was ~~then~~ generated using the summary statistics from the European ancestry meta-analysis and tested using PACG-related traits in an independent cohort, the EPIC-Norfolk Eye Study

Lines: 295-297

Remarks to the Author		Changes made in the manuscript
		The association of quintiles of PRS and PACG-related traits by quintiles of PRS-B is shown graphically in Supplementary Figure 1011. Supplementary Table 9 legend .. polygenic risk score (PRS-B), age and sex. Figure 4 legend and within the plot .. for age, sex and PRS-A (red). C Odds ratio of PACG and PAC cases versus controls by quintiles of multi-trait analysis of genome-wide association study (MTAG) PRS-A with quintile 1 as reference. Supplementary Figure 11 legend .. polygenic risk score (PRS-B) in quintiles Supplementary Figure 12 legend and within the plot .. age, sex and PRS-B (red). C. Odds ratio of PACG and PAC cases versus controls by quintiles of European meta-analysis PRS-B with quintile 1 as reference.

(b) As well as age and sex, include refractive error in the ‘baseline’ AUROC analysis for PACG, since refractive is a well-known and easily measured risk factor for PACG;

Values of the AUROC for a baseline model that includes age, sex and spherical equivalent (SE) is now given in the text. We found that the AUROC for a model that included age, sex and SE was very similar to the AUROC for our original baseline model which included just age and sex.

Additionally, while a more hypermetropic SE is significantly associated with PACG adjusted for age and sex, it is no longer significant when further adjusted for PRS-A in a logistic regression model. While our PRS is clearly in part driven by refractive error (RE), it is possible that some drivers of RE may be particularly causal for PACG and our PRS is enriched for these and not for other aspects of RE which may not predispose to PACG. This suggests that our PRS may have utility over and above a one-off measurement RE.

We have made the following changes in the “Population level prediction performance using PRS” section.

Lines: 267-273

Models additionally including spherical equivalent (SE) gave similar associations. A model including PRS-A, SE, age and sex has AUROC=0.75, while a model including SE, age and sex had AUROC=0.69 (Δ AUROC 0.06, $P_{DeLong} 2.32 \times 10^{-2}$). The association of PACG across quintiles of PRS-A ($P_{trend} = 9.89 \times 10^{-5}$) and Q1 versus Q5 of standardised PRS-A (OR 7.80 (95% CI 2.18-49.90)) for these models adjusted for SE, age and sex (Supplementary Figure 10) were similar to equivalent models adjusted for age and sex only (Figure 4).

Lines: 273-279

A more hypermetropic SE was significantly associated with PACG in a logistic regression model adjusted for age and sex (OR per dioptre 1.24, 95% CI 1.07-1.44; $P=4.77 \times 10^{-3}$) (Supplementary Table 8). Further adding PRS-A to this model substantially improved the pseudo R^2 (from 3.8% to 5.9%); PRS-A, after standardisation to mean=0 and standard deviation=1, was significantly associated with PACG in this model (OR 1.79, 95% CI 1.32-2.42; $P=1.83 \times 10^{-4}$) but

refractive error was no longer significant (OR 1.13, 95% CI 0.97-1.32; $P=0.12$).

Lines: 297-307

PACG/PAC associations for models using PRS-B were weaker than those using PRS-A. An AUROC=0.71 for a model including PRS-B, SE, age and sex was non-significantly greater than AUROC=0.69 for a model for SE, age and sex (Δ AUROC 0.02, P_{DeLong} 0.26).

The top quintile of standardised PRS-B was significantly associated with PACG compared to the bottom quintile, adjusted for SE, age and sex, OR 2.26 (95% CI 1.01-5.54). The trend across quintiles was also significant ($P_{trend} 2.54 \times 10^{-2}$). Similarly, an AUROC=0.69 for a model including PRS-B, age and sex was not significantly different to an AUROC=0.67 for a model with just age and sex (Δ AUROC 0.02, P_{DeLong} 0.43) while the association across quintiles of PRS-B was $P_{trend} = 0.023$ and the association for Q1 versus Q5 of standardised PRS-B was OR 2.41 (95%CI 1.08-5.90) (Supplementary Figure 12).

An additional figure has been added to the Supplementary Figures.

Supplementary Table 8: Logistic regression models

Remarks to the Author		Changes made in the manuscript
(c) report delta-AUROC values instead of raw AUROC values, where delta-AUROC is the AUROC for the PRS + Age + Sex + refractive error minus the AUROC for the Age + Sex + refractive error: delta-AUROC = AUROC [PRS + Age + Sex + refractive error] - AUROC [Age + Sex + refractive error]	We have also added delta AUROC (shown as ΔAUROC), the difference between baseline models and those including PRS, to the text when comparing models.	We have made the following changes in the “Population level prediction performance using PRS” section. Lines: 261-264 The area under the receiver operating characteristic curve (AUROC) for a model including PRS-A, age and sex was 0.75, while a model including age and sex alone had AUROC=0.67, (difference in AUROC between models (ΔAUROC) 0.08, $P_{DeLong} 9.49 \times 10^{-3}$). Lines: 267-269 Models additionally including spherical equivalent (SE) gave similar associations. A model including PRS-A, SE, age and sex has AUROC=0.75, while a model including SE, age and sex had AUROC=0.69 (ΔAUROC 0.06, $P_{DeLong} 2.32 \times 10^{-2}$). Lines: 335-337 Using a model that included PRS-A and sex gave an AUROC of 0.63, significantly greater than the AUROC for sex alone of 0.57, (ΔAUROC 0.06, $P_{DeLong} 1.66 \times 10^{-2}$)

Remarks to the Author		Changes made in the manuscript
(d) Provide the AUROC result for PACG in the EPIC-Norfolk Eye Study cohort first (line 247), before reporting the results for other traits;	We have reworked this section based on the suggestions made by the reviewers and reordered the results as requested. See also (a), (b) and (c) above.	We have made the following changes in the “Population level prediction performance using PRS” section. Lines: 254-279 Lines: 280-307 Please see above.

(e) At the conclusion of this section and/or in the conclusion of the manuscript, explain

(i) how much better the PRS would need to be for use clinically,

Our study has demonstrated the potential clinical utility of our PRS after accounting for SE. We have now included logistic regression models (Supplementary Table 10) that further illustrate the predictive power of the PRS. In addition, we have performed GWAS sample size simulations using GENESIS. These results suggest that increasing GWAS sample size corresponds to an increase in variance explained and better performing PRSs. The PRS is already showing potential for utility over and above SE, and with modest, achievable improvements in its performance it may start to be useful in a clinical setting.

We have made the following changes in the “Population level prediction performance using PRS” section.

Lines: 273-279

Please see above (text on additional logistic regression analysis demonstrating predictive utility over and above SE).

The following sentence was added to a new section “Strengths and Limitations”.

Line: 373-376

While our results suggest potential clinical utility for our PRS over and above a measurement of SE, our validation samples are small, and our simulations suggest that larger GWAS in the future will generate better performing PRSs (Supplementary Figure 14).

An additional table has been added to the Supplementary Tables.

Supplementary Table 8: Logistic regression models

An additional figure has been added to the Supplementary Figures.

Supplementary Figure 14: “Simulations of GWAS sample size by number of discovered variants, AUROC and variance explained using the PACG European meta-analysis”

(ii) the potential sub-optimal performance of the PRS in non-EUR ancestries due to trans-ancestry portability, and the ethical impact this may have on clinical implementation;

We have now carried out further analyses of the performance of our PRS in participants of non-European ancestry that were excluded during the selection of our European cohort. Our results support the predictive utility of the PRS in participants of non-European ancestry, though we acknowledge that further work is needed in larger and more diverse studies before this approach can be implemented clinically.

We have added a sentence to the “Imputation” section in the Methods.

Lines: 453-455

Non-European participants were separately imputed using the 1000 Genomes phase 3, v.5 panel, consisting of genomes of 2,504 individuals from 26 populations on the Michigan Imputation Server.

We have added text to the PRS sections in the Results and Discussion.

Lines: 333-342

We also tested the performance of our PRS in participants of non-European ancestry, 137 PACG and acute primary angle closure (APAC) cases and 245 controls, that were excluded during the selection of our European cohort. Using a model that included PRS-A and sex gave an AUROC of 0.63, significantly greater than the AUROC for sex with alone of 0.57, (Δ AUROC 0.06, $P_{\text{DeLong}} 1.66 \times 10^{-2}$) and there was an association with PACG/APAC over quintiles of PRS-A ($P_{\text{trend}} = 1.07 \times 10^{-3}$) (Supplementary Figure 13). We did not include age in these analyses as this was not available for these cohorts. Our results support the predictive utility of the PRS in participants of non-European ancestry, though we acknowledge that further work is needed in larger and more diverse

Remarks to the Author		Changes made in the manuscript
		studies before this approach can be implemented clinically. An additional figure has been added to the Supplementary Figures. Supplementary Figure 13: “Polygenic risk score PRS-A from the MTAG analysis applied to 137 PACG and APAC cases and 245 controls from replication cohort participants with non-European ancestry”

Remarks to the Author		Changes made in the manuscript
(iii) estimate how many more PACG cases will need to be recruited to increase the accuracy of the PRS sufficiently for clinical use	We have used GENESIS to estimate the sample size needed to use PRS for clinical use. See comments above.	An additional figure has been added to the Supplementary Figures. Supplementary Figure 14: “Simulations of GWAS sample size by number of discovered variants, AUROC and variance explained using the PACG European meta-analysis”
2. GWAS in UK Biobank. (a) The methods section reported that only 509,485 were studied in the UK Biobank GWAS for PACG. Was this a typo, or were only directly-genotyped variants studied? If the latter, why use directly-genotyped variants for UK Biobank but imputed variants for the other cohorts?	We mistakenly quoted the number of directly genotyped variants, taken from the log of the first stage of GWAS processing (Regenie Step 1) which uses only directly called genotypes. However, imputed variants were used for GWAS discovery and replication and the correct figure for post-QC variants in UK Biobank is now shown, using the output summary statistics. The figure is also now quoted in the study diagram (Figure 1)	We have edited the following sentence in the Genome-Wide Association Study section. Lines 463-464 Total participants prior to QC were 488,371 and after QC were 440,611 while 509,485 11,234,784 variants remained after QC. We have modified the Study Diagram. Figure 1: Discovery cohort (European ancestry), UK Biobank Post QC Variants: 11,234,784.

Remarks to the Author		Changes made in the manuscript
(b) One of the genome-wide significant variants (PCDH7) had a MAF=0.01 and did not replicate. Such false positives are common in a GWAS with a small number of cases, as was true here (n=1564 cases). What was the statistical power to detect (e.g. an OR>2 or an OR<0.5) a genome-wide significant variant for MAFs <0.05? The authors may wish to choose a more stringent MAF threshold for the discovery GWAS, to reduce the risk of false positives.	We agree that false positives are to be expected in studies such as this one. To address this, our study design involves the use of several replication cohorts to help identify potential false positives. The choice of a minor allele frequency threshold is arbitrary and making it more stringent may also eliminate true positives. Additionally, we agree with the reviewer’s important point regarding the relatively small number of cases and now highlight the potential for false positives in the manuscript.	We have added the following to the new “Strengths and Limitations” section. Lines 355-360 PACG is relatively uncommon in Europeans and despite our discovery cohort having >500,000 participants, case numbers were modest. Our choice of MAF<0.01 was arbitrary, but attempted to balance minimising false positives without eliminating true positives. While false positives such as PCDH7 were not unexpected, the use of several replication cohorts helped identify them.
3. Mendelian randomization. (a) When testing for a causal effect of refractive error on PACG, it would be more informative to use GWAS summary statistics for refractive error in units of diopters. The current MR effect estimate is expressed as an OR per unit increase in a GWAS Z-score for refractive error, which is not easy to interpret. An MR effect estimate in units of OR per D would be easily understood.	We use the largest GWAS for refractive error, (Hysi et al., Nat Genet 2020), in order to use the most accurate information to examine causality. However, this study in part used refractive error inferred from questionnaire data and therefore not on a diopetre scale. Therefore, the final results from this study were on a Z-score scale rather than a diopetre scale. We have now clarified in the Results the reason for the scale of refractive error genetic instruments.	The following additions were made to the “Mendelian randomisation” section in the Methods. Lines: 547-551 We use the largest GWAS for refractive error, (Hysi et al., Nat Genet 2020), in order to use the most accurate information to examine causality. The study reported findings using Z-scores rather than diopetres since this study in part used refractive error inferred from questionnaire data. Therefore, the final results from our MR analysis are on a Z-score scale rather than a diopetre scale.

(b) According to Supplementary Table 6, the GWAS summary statistics for the trait 'hypermetropia' were obtained from the MRC-IEU database. In my opinion, none of the available hypermetropia summary statistics in the MRC-IEU database are suitable for MR. The summary statistics either have a very small sample size, or they assumed that (UK Biobank) participants not asked about their reason for wearing glasses were unaffected, i.e. they were classified as controls. For a common trait such as hypermetropia, such massive misclassification will yield incorrect beta coefficients, which will severely bias the MR causal estimate.

Supplementary Table 6 shows genetic correlations using some routinely produced summary statistics from publicly available sources. Its purpose was to provide a broad overview of correlations with phenotypes suspected of being related to PACG. However, these summary statistics were not used for our Mendelian randomization (MR) analysis. Our MR used only summary statistics from published results where accurate measurement techniques were used.

Although questionnaire-based methods for determining hypermetropia are imperfectly defined, we found a very high correlation between hypermetropia and refractive error (RE) and a high correlation with axial length (AL) both of which were measured using more accurate techniques (Supplementary Figure 6). AL, RE and hypermetropia show similar correlations with PACG.

We have now clarified that the genetic correlation analyses with the hypermetropia was an exploratory analytical approach using publicly available data on a broad range of mainly non-ocular traits. We also clarify that the GWAS results used from our MR study are different to the ones used in the exploratory genetic correlation analyses.

We have added the following text to the Genetic correlations section.

Lines: 204-206

This exploratory analytical approach, using summary statistics from publicly available sources, provides a broad overview of phenotypes suspected of being related to PACG.

Lines: 541-543

For eye colour, a GWAS using 23andMe and the VisiGen Consortium was used (Simcoe et al., 2021). This is the largest GWAS of eye colour available; the phenotype was not measured in UK Biobank.

Lines: 547-548

We use the largest GWAS for refractive error (Hysi, et al., 2020) in order to use the most accurate information to examine causality.

(c) The exact set of GWAS summary statistics used for each MR analysis should be presented in Supplementary Table 6. For example, currently, the label 'MRC-IEU' for hypermetropia does not make it clear which of the 6 sets of hypermetropia summary statistics available in the MRC-IEU database was used.

As mentioned above, Supplementary Table 6 was not used for MR analysis. However, we agree that the table should state precisely which summary statistics were used and we have corrected this. We also clarified in the Genetic correlation section that the summary statistics were from public sources (see above) and made clear in MR section that the summary statistics were from published GWAS.

Additionally, we have now stated the sources used for our MR analysis in Supplementary Table 6 (formerly Supplementary Table 7)

We have made changes to Supplementary Table 5 (formerly Supplementary Table 6) in the "Reference" column.

Standing height "MRC-IEU: ukb-b-10787"
Waist circumference "MRC-IEU: ukb-b-9405"

Darker Skin colour "MRC-IEU: ukb-b-19560"
Black hair colour "MRC-IEU: ukb-d-1747_5"
Hypermetropia "MRC-IEU: ukb-b-18189"
Age when started wearing glasses "MRC-IEU: ukb-b-5801"

We have altered the legend of Supplementary Table 5.

Genetic correlation of the largest publicly available summary statistics for a set of traits using linkage disequilibrium score regression.

We have altered the legend of Supplementary Table 6 (formerly Supplementary Table 7)

Exposure summary statistics for axial length were from the GERA cohort (Jiang et al., 2023), eye colour from 23andMe and the VisiGen Consortium (Simcoe et al., 2021) and refractive error from UK Biobank (Hysi, et al., 2020)

Remarks to the Author		Changes made in the manuscript
4. Supplementary tables and figures. (a) It would be informative to include Supplementary Table 3 in the main text, instead of the supplement. Please add the MAF in each ancestry group to this table.	We have promoted Supplementary Table 3 to main Table 2 as requested.	We have made the following changes to the Tables. Table 2: Supplementary table 3 has been renamed as main table 2. Other supplementary tables have been renumbered accordingly.
(b) Ticks are used to indicate the level of statistical support in Supplementary Table 4. Why not simply present the exact p-value?	We have modified Supplementary table 4 and now present the exact p-value.	We have made the following changes to the Tables. Supplementary Table 3: The table now includes exact p-values.
(c) Ticks are used to indicate prior reports of association in Supplementary Table 5. Instead, I recommend citing the relevant publication directly.	We agree with your suggestion and have added reference IDs to the table which indicate the relevant publication or dataset.	We have made the following changes to the Tables. Supplementary Table 4: The table now includes Open Targets Genetics IDs which indicate the relevant publication or dataset.

Remarks to the Author		Changes made in the manuscript
(d) Define the error bars in Supplementary Figure 4.	The definition of the error bars has been added to the legend on Supplementary Figure 4 and to other figures that use error bars.	We have made the following changes figures legends. Supplementary Figure 4: Error bars represent 95% confidence intervals. Figure 3: Error bars represent 95% confidence intervals. Supplementary Figure 7: Error bars represent 95% confidence intervals. Supplementary Figure 8: Error bars represent 95% confidence intervals.

Remarks to the Author		Changes made in the manuscript
5. GWAS in UK Biobank and Europeans. The effect sizes of the PACG loci are strikingly high, e.g. OR > 1.2. The authors may wish to examine if these loci show any evidence of dominance or recessive association, or whether an additive model provides a better fit (see, for example, PMID: 32227305).	While we acknowledge that examination of individual loci may indicate some dominant or some recessive associations, we consider that using an a priori and consistent modelling approach is reasonable. There would also be some danger in overfitting if we attempted to use a model determined post-hoc. However, we have examined the associations per allele for the genome-wide significant associations and now present associations from additive, dominant and recessive models in a new table (Supplementary Table 10).	The following addition has been made to the Methods section. Lines: 473-478 Additive modelling was adopted a priori to ensure a consistent analytical approach and reduce the likelihood of overfitting. Per-allele genome-wide associations were examined, and dominant and recessive models are shown in Supplementary Table 10. It is possible that a recessive model better fits the association with PACG at rs10818834, given the larger effect size and smaller P-value compared to the additive model.
6. Line 154. The sentence beginning, “Regional association...” Did not link well with the preceding sentences. Could this point be made elsewhere?	We have moved the sentence mentioning the regional association analysis to a location near the start of the “European ancestry meta-analysis” section.	We have moved a sentence in the European ancestry meta-analysis section. Lines: 146-148 Regional association analysis was performed to visualise the loci identified and illustrate linkage disequilibrium between variants within each locus (Supplementary Figure 2).

Remarks to the Author	Response	Changes made in the manuscript
Reviewer #2		Lines numbers shown for tracked changes document Additional text is underlined ; deleted text is struck through.

This paper reports the first large-scale genome-wide association study (GWAS) to investigate genetic risk for Primary Angle-Closure Glaucoma (PACG) in Europeans in the UK Biobank (UKB; 1564 cases and 439,185 controls). They detected genome-wide significance at five loci, including three novel loci and two loci previously associated with PACG in Asian populations. They then examined the top associated variants in six independent replication cohorts (1619 cases and 334,031 controls from six national cohorts: Australia, Brazil, Italy, Finland, UK and USA). In meta-analysis of the replication cohorts, four variants were at least nominally significant ($P < 0.05$) and all had consistent direction of effect. To increase power to detect PACG associations, they meta-analysed all European cohorts, finding a total of 10 genome-wide significant loci. They also compared results to previously published findings in Asian-focused analyses and, to further increase power, they performed a multi-ancestry meta-analysis combining their European ancestry meta-analysis with 14 cohorts from Asian countries including China, Vietnam, Thailand, Malaysia, Singapore, Japan and India. An additional 12 novel loci reached genome-wide significance in

Thank you for your detailed comments and raising several important points which we have addressed.

Remarks to the Author	Response	Changes made in the manuscript
the multi-ancestry analysis that were not independently significant in the Asian or European ancestry meta-analyses. They performed conditional analyses to identify independent effects within each locus, genetic correlation analysis to assess shared genetic architecture between PACG and PACG-related traits, and Mendelian randomisation (MR) using multiple software packages to examine potential causal relationships between PACG and axial length, eye colour, and refractive error). They performed multi-trait analyses with MTAG and conditional analyses, and generated two polygenic risk scores (PRS) from the MTAG analysis and the European meta-analysis, both of which were tested in the independent EPIC-Norfolk Eye Study to determine association between the PRS and PACG, axial length, anterior chamber depth, spherical equivalent, blue eye colour, primary open-angle glaucoma, and intraocular pressure. This manuscript is very well written and, for the most part, clearly presented. I have a few questions, suggestions, and concerns, as detailed below, that I think will clarify and possibly improve the manuscript.		

Remarks to the Author	Response	Changes made in the manuscript
-Considering the case definition in UKB, how likely is it that an individual would have an ICD-9 or ICD-10 code for PACG but not truly have one of those diagnoses?	It is likely that a case diagnosis relying on hospital coding data will be less reliable than ascertainment based on clinical examination as part of the study. It was therefore important that our study included replication data from 6 cohorts in which the clinical ascertainment of PACG was more robust. Recent studies (Biggerstaff et al., 2016, Ophthalmic Epidemiology and Lu et al., 2024, Clinical Epidemiology) estimated sensitivity, specificity, positive predictive value and negative predictive value of PACG coded using ICD9 and ICD10 against a gold standard of direct clinical ascertainment of PACG. These studies concluded that routine clinical coding of PACG has sufficient accuracy for epidemiologic research. We have now summarised these issues in the “Strengths and Limitations” section.	We have added the following to a new section “Strengths and Limitations”. Lines: 364-370 PACG was ascertained in our discovery cohort using routine hospital medical coding. While we acknowledge that this may result in a higher degree of misclassification than a direct clinical examination as part of the study, recent studies (Biggerstaff et al., 2016, Ophthalmic Epidemiology and Lu et al., 2024, Clinical Epidemiology) have reported sufficient accuracy for epidemiologic research for PACG from ICD coding. Additionally, 6 of our replication cohorts used multiple clinical criteria for ascertainment and were not reliant on ICD codes from routine healthcare record linkage.

Remarks to the Author	Response	Changes made in the manuscript
-Given that the main paper methods only list ICD-9 and ICD-10 codes for PACG, the Supplementary Methods' mention of the following in the inclusion criteria was confusing: "Patients with acute primary angle-closure (APAC) or primary angle-closure glaucoma (PACG) were recruited," especially given that the title includes "PACG" and the main paper focuses on PACG. Further, is it typical to jointly analyze PACG with APAC patients? Please provide rationale	The study uses replication cohorts some of which use a broader clinically-based definition for the outcome which includes acute primary angle-closure (APAC) as well as primary angle-closure glaucoma (PACG). While the text in the Supplementary Note states that the definition only refers to replication cohorts, it does not make it sufficiently clear which cohorts the broader definition applies to. In particular the definition does not apply to FinnGen or to EPIC-Norfolk. This has now been corrected. We included APAC patients as they are at very high risk of developing vision impairment from PACG if left untreated (Aung et al., Ophthalmology 2004).	We made the following changes in the “Population level prediction performance using PRS” section. Lines: 333-335 We also tested the performance of our PRS in participants of non-European ancestry, 137 PACG and acute primary angle closure (APAC) cases and 245 controls, that were excluded during the selection of our European cohort. We have made the following changes to the “Replication Cohorts” section. Lines: 411-416 PACG and APAC were was clinically ascertained in replication cohorts from Italy, UK, USA, Brazil and Australia (557 cases, 7,597 controls) while the FinnGen cohort used medical coding from primary and secondary care (1,062 cases, 326,434 controls). Consistent with the previous largest GWAS for PACG (Khor et al., Nat Genet 2016, Vithana et al., Nat Genet 2012), we have included participants with APAC as they are very high risk for developing vision impairment from PACG if left untreated (Aung et al., Ophthalmology 2004).

-The Principal Components analysis (Supplementary Figure 12) and Methods describe exclusion of individuals who are not European from the replication cohorts; how likely are these findings to apply to individuals of predominantly non-European ancestry? Would it be possible to evaluate the novel loci and performance of the risk scores in those individuals who were excluded from initial analyses due to more heterogeneous ancestry, or in another ancestral group?

We are grateful for this suggestion and have performed an additional analysis using participants with non-European ancestry excluded from our European cohort selection process. We imputed variants for this group using the 1000 Genomes multi-ancestry panel and tested predictive performance using PRS-A. Despite the small number of non-Europeans, our PRS was able to demonstrate effectiveness in this group. We have summarised these findings in the Results and in a new figure.

We have added the following text to the Imputation section of the Methods.

Lines: 450-455

Non-European participants were separately imputed using the 1000 Genomes phase 3, v.5 panel, consisting of genomes of 2,504 individuals from 26 populations on the Michigan Imputation Server.

We have added a new paragraph to the “Population level prediction performance using PRS” section.

Lines: 333-342

We also tested the performance of our PRS in participants of non-European ancestry, 137 PACG and acute primary angle closure (APAC) cases and 245 controls, that were excluded during the selection of our European cohort. Using a model that included PRS-A and sex gave an AUROC of 0.63, significantly greater than the AUROC for sex with alone of 0.57, (Δ AUROC 0.06, P_{DeLong} 1.66×10^{-2}) and there was an association with PACG/APAC over quintiles of PRS-A ($P_{trend} = 1.07 \times 10^{-3}$) (Supplementary Figure 13). We did not include age in these analyses as this was not available for these cohorts. Our results support the predictive utility of the PRS in participants of non-European ancestry, though we acknowledge that further work is needed in larger and more diverse

Remarks to the Author	Response	Changes made in the manuscript
		studies before this approach can be implemented clinically. We have added a new figure. Supplementary Figure 13: “Polygenic risk score PRS-A from the MTAG analysis applied to 137 PACG and APAC cases and 245 controls from replication cohort participants with non-European ancestry”

Remarks to the Author	Response	Changes made in the manuscript
-In Supplementary Figures 10 and 11, Polygenic risk score from the European meta-analysis and spherical equivalent, axial length, anterior chamber depth and blue eye colour in 7,223 EPIC-Norfolk participants (10) and Polygenic risk score from European meta-analysis applied to 47 PACG cases and 6,623 controls in the EPIC-Norfolk study (11) are presented. What are the implications of these findings in Europeans, as well as other ancestral groups?	The results of our study point to various mechanistic causes of PACG with ocular biometry and anatomy the predominant factors. While traits such as blue eye colour differ by ancestral group, and our PRS was created from European cohorts, it was able to demonstrate effectiveness in both Europeans and Asians.	The following sentence was added to a new section “Strengths and Limitations”. Lines: 372-373 Although traits such as blue eye colour differ by ancestral group our PRS was able to demonstrate effectiveness in participants from European and Asian ancestries.
-The reference panels and imputation software differ between the different groups analyzed; please provide rationale for why this would not potentially influence/increase the likelihood of detecting spurious results (especially when some of the datasets are obviously much larger than others and clearly driving significance of signal detection).	Given that we analysed each study separately and then carried out meta-analysis as a separate step, the different imputation panels and reference studies are less likely to be an issue. Most importantly, the cases and controls from each study used the same panel. We acknowledge the potential limitation of different imputation panels in the “Results and Discussion” section.	We have added the following to a new section “Strengths and Limitations”. Lines: 360-364 Reference panels and imputation software differed between the cohorts used in our study. However, we analysed each cohort separately before meta-analysing the results. Importantly, the cases and controls from each study used the same imputation panel, thus minimizing any potential biases between cases and controls (which will drive spurious results).

Remarks to the Author	Response	Changes made in the manuscript
-Please provide justification for the imputation information score cutoff used (<0.4).	Authors of different software used for imputation recommend different cut-offs, the choice of which is in part arbitrary. Here we chose something slightly more conservative than the often recommended 0.3. Notably, the imputation scores for all our main significant loci, were higher than 0.85. The imputation information score now appears in Table 1.	We have added the following sentence to the Methods section Lines: 590-593 The choice of imputation score cut-off is arbitrary and unlikely to have a large impact on our results, given that our genome wide significant loci all had an imputation score > 0.85 (Table 1)
-Please provide a reference for "McCarthy Group Tools."	We have been unable to find a suitable reference specific to the McCarthy group tools used for pre-imputation. We have instead added a web link to the text.	The following web link has been added to the Methods section. Lines: 441-444 Pre-imputation quality control and exclusion was applied to genetic data from each replication cohort using the McCarthy Group Tools (https://www.well.ox.ac.uk/~wrayner/tools/)
-Please provide justification for use of Regenie (as opposed to other software, e.g. GENESIS, SAIGE, and fastGWA-GLMM).	Regenie was chosen for our analyses pipeline because of its strong capability for handling cryptic relatedness using Firth regression, which is suitably implemented, and given the relatedness structure in UK Biobank.	The following has been added to the Methods section. Lines: 606-609 Regenie was chosen for our analyses pipeline because of its strong capability for handling cryptic relatedness using Firth regression, which is suitably implemented, and given the relatedness structure in UK Biobank.

Remarks to the Author	Response	Changes made in the manuscript
-Figure 1 should be updated for clarity and consistency.	Thank you for this suggestion. We have modified Figure 1 to help improve readability and provide additional information. Figures have been cross-checked with those appearing in the manuscript and tables. Formatting and alignment have also been improved.	We have made changes to Figure 1.  - the numbers of participants with European and non-European ancestry by cohort - details of imputations panels by ancestry - numbers of PACG cases and controls post imputation and QC by ancestry - the number of imputed variants in the discovery cohort before and after QC - the number of PACG cases and controls in the EPIC-Norfolk cohort. We have modified the Identification of European Ancestry section in the Methods. Lines: 432-433 The number of exclusions by replication cohort were Australia 65, Brazil 267, Italy 15, UK 10 and USA 25. Participant numbers by cohort for European and non-European participants are shown in Figure 1.

Remarks to the Author	Response	Changes made in the manuscript
-Please explain the use of METAL for the multi-ancestry meta-analysis as opposed to a program developed specifically to take multi-ancestry into account (e.g. MR-MEGA).	We opted to use METAL, one of the most well-established meta-analysis software systems, because we were specifically interested in modelling fixed effects in these cohorts. Fixed effects analyses are more likely to identify loci with more consistent effects across ethnic groups. We are aware of alternative meta-analysis software packages that have been brought into the field more recently as well as the advantages they would offer to researchers interested in modelling their analyses in a different way.	We have now added the following rationale to our Methods section. Lines: 613-616 METAL is a well-established meta-analysis software system. We aimed to identify homogenous associations and therefore used a fixed effects meta-analysis package. Fixed effects analyses are more likely to identify loci with more consistent effects across ethnic groups.

Remarks to the Author	Response	Changes made in the manuscript
-Please elaborate on limitations/weaknesses of the study. E.g. in the replication meta-analysis, the authors acknowledge that the associations were largely driven by the FinnGen, UK, and Australian cohorts, but elaboration on the implications would be helpful.	We agree with the reviewer that we have not made study limitations sufficiently clear but also that study strengths have not always been stated. For this reason, we have extended the “Results and Discussion” section with a paragraph which outlines the strengths and limitations of the study. This includes areas such as the use of routine coding for discovery ascertainment (previously mentioned), small numbers of participants with non-European ancestries in UK Biobank and differing sample sizes in replication cohorts. While associations in replication cohorts were driven by the larger cohorts, this is a function of sample size rather than any inherent weakness in other cohorts.	We have added the following to a new section “Strengths and Limitations” Lines: 364-371 PACG was ascertained in our discovery cohort using routine hospital medical coding. While we acknowledge that this may result in a higher degree of misclassification than a direct clinical examination as part of the study, recent studies (Biggerstaff et al., 2016, Ophthalmic Epidemiology and Lu et al., 2024, Clinical Epidemiology) have reported sufficient accuracy for epidemiologic research for PACG from ICD coding. Additionally, 6 of our replication cohorts used multiple clinical criteria for ascertainment and were not reliant on ICD codes from routine healthcare record linkage. While associations in replication cohorts were driven by the larger cohorts, this is a function of sample size rather than any inherent weakness in other cohorts.

Remarks to the Author	Response	Changes made in the manuscript
-After reading this, I am impressed by the multitude of analyses presented and the novel (and consistently replicated) results. What I am left wondering is - what is the point? I think it would be very helpful to provide additional discussion on the value these results bring to the field.	Our study provides mechanistic insights and highlights the differences between Europeans and Asians. It also affirms the known clinical associations with PACG. Summary statistics from the study may be used in future studies for MR, genetic correlations, meta-GWAS analyses and improved polygenic risk score (PRS) performance. The study's strengths are now described in a new Strengths and Limitations paragraph in the discussion.	We have now added the following rationale to the new "Strengths and Limitations" section. Lines: 344-354 Our results provide further evidence of explainable heritability of PACG. While the association between refractive error and PACG has previously been clinically established, here we show that a genetic predisposition to a smaller eye confers risk for PACG. Importantly, we demonstrate a significant association of our PRS with PACG even when including refractive error. We also demonstrate genetic factors related to iris morphology as increasing risk for PACG, although previous studies have reported only modest improvements in PACG detection of genetic factors over anterior segment imaging parameters (Nongpiur et al. 2019). While our current GWAS are underpowered to derive an accurate PRS, we hypothesise that larger discovery sample sizes in the future together with leveraging genetic correlation with refractive error, will enable the development of clinically useful PRS for risk of PACG.

Remarks to the Author	Response	Changes made in the manuscript
Reviewer #3		Lines numbers shown for tracked changes document Additional text is underlined ; deleted text is struck through.

This is an excellent and very interesting paper which describes the results of a Genome-Wide Association study for primary angle-closure glaucoma (PACG), and identified novel loci related to ocular biometry and morphology.

Although previous work has been conducted on primarily Asian populations, where PACG is more common, this new study includes two large European cohorts, the UK Biobank and the EPIC-Norfolk studies.

The most significant associations were HERC2/OCA2 (rs12913832:G) previously associated with brown iris colour and a second locus near SEMA3A (rs17245595:C) which has been associated with morphological characteristics and colour of the iris. The morphology previously published was iris crypts but this is not mentioned in the paper.

The authors also identified loci at PXDNL (rs11984688:A), and GLIS3 (rs746970:G), both previously associated with PACG in Asian populations), and at PCDH7 (rs184176302:C)

Six novel loci identified by meta-analysis included three genes previously associated with refractive error: LAMA2 (rs12193446:A), KCNQ5 (rs7744813:C) and DENND1A (rs10818834:T), Mendelian Randomisation confirmed the

Thank you for these comments and the acknowledgement that the paper advances the understanding of PACG.

Remarks to the Author	Response	Changes made in the manuscript
association of shorter axial length and hyperopia with PACG. Polygenic Risk Score showed association with shallower anterior chamber and higher intraocular pressure. This is an important advance in the understanding of PACG.		

Remarks to the Author	Response	Changes made in the manuscript
Suggestions The authors claim this is the FIRST large scale GWAS of European PACG with 1,564 cases, but the previous study (Khor 2016 including some of the current authors), which although predominantly Asian included 1,105 European PACG cases which were used in the replication of the current study. Thus the authors should change the claim of primacy.	We agree with the reviewer and acknowledge that Khor et. al. used a large group of participants with European ancestry. When we referred to our study being the first large scale GWAS in Europeans, we meant that it was the first to use a European cohort for discovery. We have clarified this point in the text to make this distinction clear.	We have made the following changes to the Abstract, and Introduction. Lines: 50-51 We carried out the first large-scale discovery GWAS for PACG in Europeans using the UK Biobank Lines: 99-101 Here we report the first large-scale GWAS for PACG in Europeans using a European discovery cohort, using UK Biobank (UKB), with replication in six independent European populations
The authors should mention that GLIS3 has also been associated with Congenital Glaucoma and with POAG.	Although we listed GLIS3 in Supplementary Table 5 as associated with ‘any form of glaucoma’, we were not specific about the forms of glaucoma. To address this, we have now made this point explicitly by stating that GLIS3 has been associated with the other forms of glaucoma mentioned by the reviewer.	We have made the following addition to the “Discovery GWAS” section. Line: 131-132 GLIS3 has also been associated with congenital glaucoma and with POAG.

The major changes I recommend to the discussion are given below:

I was stunned the authors completely missed all the previous literature linking the phenotypes of HERC2/OCA2 and SEMA3A to PACG, especially when some were involved in these earlier papers.

The previous SEMA3A association was with iris crypt frequency (Larsson 2011).

One of the senior authors from Singapore is on another paper where they correlated iris crypts with Primary Angle Closure Glaucoma.

Koh V, Chua J, Shi Y, Thakku SG, Lee R, Nongpiur ME, Baskaran M, Kumar RS, Perera S, Aung T, Cheng CY. Association of iris crypts with acute primary angle closure. Br J Ophthalmol. 2017 Oct;101(10):1318-1322.

The Singaporean group also documented “Lighter coloured irides with more crypts and/or more furrows were also associated with less convexity.”

Tun TA, Chua J, Shi Y, Sidhartha E, Thakku SG, Shei W, Tan MC, Quah JH, Aung T, Cheng CY. Association of iris surface features with iris parameters assessed by swept-source optical coherence tomography in Asian eyes. Br J

Thank you for highlighting this interesting aspect of the analysis and the extensive literature published over many decades on this topic. We realise that we didn't give sufficient prominence to this area in the manuscript. While Larsson 2011 and Sidhartha 2014 were cited in the original text along with other related literature (references 12 to 19), we did not communicate their implications sufficiently clearly. To address this, we have now added two of the additional suggested references and clarified and extended the text.

We have also modified the wording of the statement that suggested variation in iris colour was only detectable in European populations.

The following changes were made in the Abstract.

Lines: 73-74

Additionally, the effect of variation in iris colour is more readily ~~may only be~~ detectable in European populations where colour is more heterogenous.

The following additions were made in the “Results and Discussion” section.

Lines: 122-126

The relationship between lighter coloured eyes and crypts and furrows is well-established. Several studies have reported associations between iris surface features, lighter eye colour and iris thickness in Asian populations suggestive of a link with angle closure (Koh, et al., 2017, Chua, et al., 2016). However, previous GWAS of PACG were not able to detect loci related to iris crypts.

Ophthalmol. 2016 Dec;100(12):1682-1685.

“irises with more crypts and lighter color were associated with wider angle”

Sidhartha E, Nongpiur ME, Cheung CY, He M, Wong TY, Aung T, Cheng CY. Relationship between iris surface features and angle width in Asian eyes. Invest Ophthalmol Vis Sci. 2014 Oct 23;55(12):8144-8.

Thus the authors statement on line 74 of the abstract is not correct and needs to be changed as variation in eye colour can be seen in Asians

“Additionally, the effect of variation in iris colour may only be detectable in European populations where colour is more heterogenous.”

“Although few Chinese persons have multiple crypts on their irides, irides with more crypts were significantly thinner and lost more volume on pupil dilation. In view that the latter feature is known to be protective for acute angle-closure”

Chua J, Thakku SG, Tun TA, Nongpiur ME, Tan MC, Girard MJ, Wong TY, Quah JH, Aung T, Cheng CY. Iris Crypts Influence Dynamic Changes of Iris Volume. Ophthalmology. 2016 Oct;123(10):2077-84.

The issue of iris crypts and glaucoma needs to be

discussed in much more detail.

Whilst iris structural abnormalities are well known in Anterior Segment Dysgenesis and Rieger Syndrome, both of which are associated with open angle glaucoma, there are some other syndromes that can be discussed such as:

Iris ectropion syndrome (Ectropion uveae) is associated with open angle glaucoma

“The posterior pigment epithelium of the iris is found on the anterior stroma at birth. The iris surface is glassy smooth and devoid of crypts.”

Sridhar U, Tripathy K. Iris Ectropion Syndrome. 2023 Aug 25. In: StatPearls [Internet]. Treasure Island (FL): StatPearls Publishing; 2024 Jan–.

“Anterior segment abnormalities included absent iris crypts” is a feature of Knobloch Syndrome (COL18A1)

Hull S, Arno G, Ku CA, Ge Z, Waseem N, Chandra A, Webster AR, Robson AG, Michaelides M, Weleber RG, Davagnanam I, Chen R, Holder GE, Pennesi ME, Moore AT. Molecular and Clinical Findings in Patients With Knobloch Syndrome. JAMA Ophthalmol. 2016 Jul 1;134(7):753-62.

In Nanophthalmic eyes, a syndrome closer to PACG where small eyes often develop glaucoma.

“The anterior surface of the iris seemed

Remarks to the Author	Response	Changes made in the manuscript
"smoother" in nanophthalmic eyes than in normal eyes." Lu Q, He W, Lu Y, Zhu X. Morphological features of anterior segment: factors influencing intraocular pressure after cataract surgery in nanophthalmos. Eye Vis (Lond). 2020 Sep 9;7:47. However, the earliest description of the association of iris crypts and narrow angle glaucoma appears to be from 1954. Posner looking at patients with unilateral narrow angle glaucoma noted the eyes with glaucoma had a smooth stroma, while the eyes without glaucoma had well circumscribed iris crypts. Posner A. Role of peripheral iris crypts in protection from narrow angle glaucoma. Eye Ear Nose Throat Mon. 1954 Jul;33(7):427-8; And also Posner A. Peripheral iris crypts in relation to medical and surgical therapy of narrow-angle glaucoma. Am J Ophthalmol. 1955 Oct;40(4):469-74. The authors might consider reviewing the other iris crypt associated genes from the Larsson 2011 paper in their PACG cohort.		

Remarks to the Author	Response	Changes made in the manuscript
Minor comments: The authors might point out that eye colour was not available from the UK Biobank.	We agree that it is helpful to point this out and have now acknowledged that eye colour was not measured in UK Biobank.	The following was added to the “Mendelian Randomisation” section in the Methods. Line: 541-543 For eye colour, a GWAS using 23andMe and the VisiGen Consortium was used (Simcoe et al., 2021). This is the largest GWAS of eye colour available; the phenotype was not measured in UK Biobank.
Line 93, “Three quarters of the global disease burden is estimated to occur in Asia”. Suggesting adding “of PACG”.	The sentence has been clarified as suggested.	The following change was made in the Introduction. Line: 94 Three-quarters of the global disease burden of PACG is estimated to occur in Asia.
Lines 262-274 and 280-284: The authors show that adding spherical equivalent refraction (SE) to a classification model including age, sex and PRS slightly improves detection of PACG from AUROC 0.69 to 0.75 but suggest “these would be insufficiently powerful to improve risk models”. Given that SE is easier and cheaper to measure than a PRS, a relevant question may be does age + sex + SE + PRS perform better than age + sex + SE?	We have reworked the section of the results that describe polygenic risk scores and AUROCs based on your comments and the comments of other reviewers who raised similar points. The important question of whether age, sex, SE and PRS perform better than age, sex and SE alone has now been addressed. We now quote the AUROCs for each model in the results section as well as the difference in AUROCs between base models and models that also include PRS.	We have made the following changes in the section “Population level prediction performance using PRS”. Lines: 256-307 (described in detail above, in response to point 1(b) from Reviewer #1)

Remarks to the Author	Response	Changes made in the manuscript
Line 294-295: “PRS...predict a substantial proportion of angle-closure disease” It would be better to quantify “substantial” here by providing sensitivity and specificity estimates and comparing to age/sex/SE as relevant.	We agree that the phrasing used was not sufficiently precise. We have modified the Conclusion to clarify this point. The PRS section has been reworked and now includes models that include SE. In order to quantify the level of predictive performance of the PRS we also state the difference in AUROC between models with and without the PRS. We now quantify the proportion of PACG that is explained by our PRS using logistic regression.	We have made the following changes in the section “Population level prediction performance using PRS”. Lines 256-294 (described in detail above, in response to point 1(b) from Reviewer #1) Lines: 273-276 (described in detail above, in response to point 1(b) from Reviewer #1, reporting proportion of variance explained in logistic regression modelling of PACG.) We have added some text to the Conclusion. Line: 385-388 PRS derived from our results predict a substantial proportion of angle-closure disease in an independent population more strongly than refractive error, opening up the possibility for targeted screening efforts for this blinding disease in the future.

Remarks to the Author	Response	Changes made in the manuscript
References 40 and 42 are duplicates	The duplicate reference has been removed.	The following citation was removed from the References section. 42. — Kurki, M. I. et al. FinnGen provides genetic insights from a well-phenotyped isolated population. Nature 613, 508–518 (2023).
References 11 and 61 are duplicates	The duplicate reference has been removed.	The following citation was removed from the reference list. 61. — Khawaja, A. P. et al. The EPIC Norfolk Eye Study: rationale, methods and a cross-sectional analysis of visual impairment in a population-based cohort. BMJ Open 3, (2013).
The replication cohort included six national cohorts from Australia, Brazil, Finland, Italy, UK and USA. Whilst the new cohorts from Brazil and Italy include authors from those countries and the FinnGen is a publicly available resource, there are no Australian or US authors on the current paper. Are those GWAS available from the earlier Khor 2016 paper (1,105 European cases)?	Data was provided by Khor et al and an invitation for authorship was extended to all replication cohorts.	

Remarks to the Author	Response	Changes made in the manuscript
Reviewer #4		Lines numbers shown for tracked changes document Additional text is underlined ; deleted text is struck through .
I co-reviewed this manuscript with one of the reviewers who provided the listed reports. This is part of the Nature Communications initiative to facilitate training in peer review and to provide appropriate recognition for Early Career Researchers who co-review manuscripts.	Thank you for your support and contribution.